# Non-canonical antagonism of PI3K by the kinase Itpkb delays thymocyte β-selection and renders it Notch-dependent

Luise Westernberg[1†], Claire Conche[1†], Yina Hsing Huang[2,3], Stephanie Rigaud[1], Yisong Deng[1], Sabine Siegemund[1], Sayak Mukherjee[4,5,6], Lyn'Al Nosaka[1], Jayajit Das[4,5,6], Karsten Sauer[1,7*]

[1]Department of Immunology and Microbial Science, The Scripps Research Institute, La Jolla, United States; [2]Department of Pathology, Geisel School of Medicine, Lebanon, United States; [3]Departments of Microbiology and Immunology, Geisel School of Medicine, Lebanon, United States; [4]Department of Pediatrics, The Ohio State University, Columbus, United States; [5]Department of Physics, The Ohio State University, Columbus, United States; [6]Battelle Center for Mathematical Medicine, The Ohio State University, Columbus, United States; [7]Department of Cell and Molecular Biology, The Scripps Research Institute, La Jolla, United States

*For correspondence: ksauer@scripps.edu

[†]These authors contributed equally to this work

Competing interests: The authors declare that no competing interests exist.

**Abstract** β-selection is the most pivotal event determining αβ T cell fate. Here, surface-expression of a pre-T cell receptor (pre-TCR) induces thymocyte metabolic activation, proliferation, survival and differentiation. Besides the pre-TCR, β-selection also requires co-stimulatory signals from Notch receptors - key cell fate determinants in eukaryotes. Here, we show that this Notch-dependence is established through antagonistic signaling by the pre-TCR/Notch effector, phosphoinositide 3-kinase (PI3K), and by inositol-trisphosphate 3-kinase B (Itpkb). Canonically, PI3K is counteracted by the lipid-phosphatases Pten and Inpp5d/SHIP-1. In contrast, Itpkb dampens pre-TCR induced PI3K/Akt signaling by producing $IP_4$, a soluble antagonist of the Akt-activating PI3K-product $PIP_3$. $Itpkb^{-/-}$ thymocytes are pre-TCR hyperresponsive, hyperactivate Akt, downstream mTOR and metabolism, undergo an accelerated β-selection and can develop to $CD4^+CD8^+$ cells without Notch. This is reversed by inhibition of Akt, mTOR or glucose metabolism. Thus, non-canonical PI3K-antagonism by Itpkb restricts pre-TCR induced metabolic activation to enforce coincidence-detection of pre-TCR expression and Notch-engagement.

## Introduction

To generate a diverse T cell repertoire reactive against many pathogens, the T cell receptor (TCR) α and β chain genes somatically rearrange in developing thymocytes. TCR functionality is then assessed at various checkpoints. Thymocytes develop from bone marrow (BM) progenitors through successive $CD4^-CD8^-$ 'double-negative' $CD44^+CD25^-$c-$Kit^+$ DN1, $CD44^+CD25^+$c-$Kit^+$ DN2, $HSA^{high}$c-$Kit^-CD44^-CD25^+$ DN3 and $HSA^{high}CD44^-CD25^-$ DN4 stages (*Petrie and Zuniga-Pflucker, 2007*; *Xiong et al., 2011*). Productive rearrangement of one *TCRβ*-allele causes surface-expression of a pre-TCR comprised of TCRβ, pre-TCRα and signal-transducing CD3 subunits on DN3 cells (*Aifantis et al., 2006*). At the first checkpoint, β-selection, ligand-independent pre-TCR signaling triggers DN3 cell metabolic activation, proliferation and survival. It also triggers allelic exclusion of the second *TCRβ* allele, initiation of *TCRα* gene-rearrangements and differentiation via $CD8^+$-$HSA^{high}TCRβ^{low}$ immature single positive (ISP) precursors into $CD4^+CD8^+$ 'double-positive' (DP) cells (*Petrie and Zuniga-Pflucker, 2007*; *Xiong et al., 2011*). β-selection ensures that only DN3 cells

**eLife digest** T cells defend our body against cancer and infectious agents such as viruses. However, they can also cause rheumatoid arthritis and other autoimmune diseases by attacking healthy tissue. T cells recognize target cells via receptor proteins on their surface. To maximize the variety of infections and cancers our immune system can recognize, we generate millions of T cells with different T cell receptors every day.

To ensure T cells work correctly, T cell receptors are tested at various checkpoints. The first checkpoint involves a process called beta (β) selection, during which T cells produce their first T cell receptor – the so-called pre-T cell receptor. This receptor causes T cells to divide and mature, and sets their future identity or "fate". To complete β-selection, T cells must also receive signals from another surface receptor – one that belongs to the Notch family, which determines cell fate in many different tissues.

The Notch receptor and the pre-T cell receptor both activate an enzyme called PI3K – a key mediator of β-selection. But the pre-T cell receptor also activates another enzyme called Itpkb that is required for T cell development. Westernberg, Conche et al. have now investigated how these different proteins and signaling processes work and interact during β-selection, using mice that lack several immune genes, including the gene that produces Itpkb.

The results of the experiments show that during β-selection, Itpkb limits the ability of PI3K to activate some of its key target proteins. This "dampened" PI3K signaling ensures that both the pre-T cell receptor and the Notch receptor must be activated to trigger T cell maturation. Without Itpkb, β-selection can occur in the absence of Notch signaling.

As Notch signaling is important for determining the fate of many different cell types, Westernberg, Conche et al.'s findings raise the possibility that Itpkb might also regulate cell fate determination in other tissues. Moreover, Itpkb may suppress tumor development, because excessive PI3K signaling drives many cancers.

expressing a functional TCRβ chain develop further. It is the major cell-fate determining event for αβ T cells. Defective β-selection causes a DN3 block and severe immunodeficiency (*Juntilla and Koretzky, 2008*; *Aifantis et al., 2006*).

pre-TCR signaling alone is insufficient for DN-to-DP cell differentiation without co-stimulation by thymic microenvironmental signals. In particular, ligand engagement of Notch on DN3/DN4 cells promotes nutrient receptor expression, glucose uptake, metabolism, growth, survival, proliferation and differentiation. But excessive Notch signaling causes thymocyte transformation and T cell acute lymphoblastic leukemia (T-ALL). This is augmented by pre-TCR signals (*Ciofani et al., 2004*; *Ciofani and Zuniga-Pflucker, 2005*; *Campese et al., 2006*; *Fayard et al., 2010*; *Taghon et al., 2006*; *Aifantis et al., 2006*; *Tussiwand et al., 2011*). So, pre-TCR/Notch costimulation needs to be limited and elucidating the underlying mechanisms is of great importance.

Both pre-TCR and Notch activate phosphatidylinositol 3-kinases (PI3K) (*Ciofani and Zuniga-Pflucker, 2005*; *Juntilla and Koretzky, 2008*; *Fayard et al., 2010*). PI3K phosphorylate the membrane lipid phosphatidylinositol(4,5)bisphosphate (PIP$_2$) into phosphatidylinositol(3,4,5)trisphosphate (PIP$_3$). PIP$_3$ recruits and activates Itk/Tec-, Pdk1-, and Akt-family kinases by binding to their PH domains. PI3K are essential and rate-limiting for β-selection by promoting metabolism, proliferation, survival and differentiation (*Juntilla and Koretzky, 2008*; *Fayard et al., 2010*). Itk promotes activation of phospholipase-Cγ1 (PLCγ1). PLCγ1 hydrolyzes PIP$_2$ into the second messengers inositol (1,4,5)trisphosphate (IP$_3$) and diacylglycerol (DAG), which then convey downstream signals (*Aifantis et al., 2006*). *Itk* loss only subtly impairs β-selection (*Lucas et al., 2007*). Pdk1 is required for DN3/DN4 cell differentiation mostly by activating Akt, and for thymocyte proliferation through other effectors (*Kelly et al., 2007*; *Fayard et al., 2010*). Akt kinases are required for β-selection by promoting DN3/DN4 cell glucose uptake, glycolysis, viability and differentiation (*Juntilla et al., 2007*; *Fayard et al., 2007*; *Mao et al., 2007*; *Fayard et al., 2010*). Recent studies suggest important roles for the Akt activator mTORC2 and possibly the Akt downstream-effector mTORC1 in β-selection (*Lee et al., 2012*; *Tang et al., 2012*; *Chou et al., 2014*).

Canonically, PI3K function is limited through $PIP_3$-removal by the lipid-phosphatases Inpp5d/SHIP1 and Pten (*Juntilla and Koretzky, 2008*; *Fayard et al., 2010*). $Inpp5d/SHIP1^{-/-}$ early thymocytes develop normally (*Kashiwada et al., 2006*). Conditionally $Pten^{-/-}$ DN cells show constitutively active Akt and accelerated development to DP cells. They can generate DP cells without pre-TCR or Notch-signaling (*Hagenbeek et al., 2004*; *Kelly et al., 2007*; *Shiroki et al., 2007*; *Wong et al., 2012*; *Hagenbeek et al., 2014*). Notch may promote DN3/DN4 cell survival and differentiation in part by repressing *Pten* (*Wong et al., 2012*). So, limiting PI3K signaling is required for β-selection and its dependence on both pre-TCR and Notch. But many details about how pre-TCR and Notch cross-talk via PI3K are controversial, and it remains unclear why pre-TCR signaling alone is insufficient for β-selection (*Juntilla and Koretzky, 2008*; *Fayard et al., 2010*; *Hagenbeek et al., 2014*).

$IP_3$ is well known to mobilize $Ca^{2+}$ but can also be phosphorylated into inositol(1,3,4,5)tetrakisphosphate ($IP_4$) by four mammalian $IP_3$ 3-kinases (*Sauer and Cooke, 2010*). Among these, we and others have identified Itpkb as an essential TCR effector. Thymocyte development in $Itpkb^{-/-}$ mice is blocked at the DP stage due to defective positive selection (*Huang et al., 2007*; *Pouillon et al., 2003*; *Wen et al., 2004*). In thymocytes, TCR signaling activates Itpkb to produce $IP_4$, a soluble analog of the PH domain binding moiety of $PIP_3$. $Itpkb^{-/-}$ thymocytes have strongly reduced $IP_3$ 3-kinase activity and $IP_4$ levels, but normal $IP_3$ levels and $Ca^{2+}$ mobilization (*Pouillon et al., 2003*; *Wen et al., 2004*). $IP_4$ can bind to PH domains and control $PIP_3$ binding (*Huang et al., 2007*; *Jia et al., 2007*). In NK cells, myeloid cells and hematopoietic stem cells (HSC), $IP_4$ competitively limits $PIP_3$-binding to, and activation of Akt (*Jia et al., 2008*; *2007*; *Sauer et al., 2013*; *Siegemund et al., 2015*). Thus, besides $PIP_3$-turnover by Inpp5d/SHIP1 and Pten, $IP_3$ 3-kinases can limit PI3K function through a non-canonical mechanism, $IP_4$ antagonism with $PIP_3$.

Here, we present data which suggest that this non-canonical mechanism restricts pre-TCR induced pro-metabolic PI3K/Akt signaling to limit the kinetics and enforce the Notch-dependence of β-selection. $Itpkb^{-/-}$ DN3 cells were pre-TCR hyperresponsive with Akt/mTOR hyperactivation and evidence for metabolic hyperactivity. They showed an accelerated and Notch independent, but pre-TCR dependent differentiation to the DP stage. Pharmacologic inhibition of Akt, mTOR or glucose metabolism restored wildtype (WT) developmental kinetics and Notch-dependence of $Itpkb^{-/-}$ DN3 cells.

## Results

### Altered β-selection in *Itpkb*[-/-] mice

DN3 cells from $Itpkb^{+/+}Rag2^{-/-}$ but not $Itpkb^{-/-}Rag2^{-/-}$ mice express Itpkb (*Figure 1*). To study if Itpkb is required for DN3 cell development, we analyzed DN cell subsets in $Itpkb^{+/+}$ (WT) vs. $Itpkb^{-/-}$ mice by flow-cytometry. For enhanced sensitivity, we gated out lineage-marker positive ($Lin^+$) non-T cells, γδ T cells and $HSA^{low}$ mature DN αβ T cells (*Bruno et al., 1996*). Compared to controls, $Itpkb^{-/-}$ mice had increased DN3 and reduced DN4 cell proportions with a ~three-fold increased DN3:DN4 ratio (*Figure 2A,B*). Blocked β-selection usually increases this ratio via accumulation of pre-selection DN3 cells and loss of DN4 cells and descendants, ultimately reducing thymic cellularity (*Michie and Zuniga-Pflucker, 2002*). Surprisingly, $Itpkb^{-/-}$ mice had WT-like total thymic cellularity and numbers of DN3 cells and $CD25^{int}$ 'DN3-4' intermediates between DN3 and DN4 cells (*Xiong et al., 2011*) (*Figure 2C*).

$Itpkb^{-/-}$ DN3, DN3-4 and DN4 cells contained WT amounts of total TCRβ and CD3 protein (*Figure 2D*). Upon successful TCRβ-rearrangement, DN3 cells express intracellular TCRβ protein which is then transported to the cell surface (*Aifantis et al., 2006*). Due to constitutive endocytosis, only small surface TCRβ/CD3 amounts are detectable on DN3 and DN3-4 cells (*Panigada et al., 2002*). These were similar between genotypes (*Figure 2D*). In contrast, the proportion of $Lin^-HSA^{high}$ DN4 cells expressing surface TCRβ/CD3 was reduced in $Itpkb^{-/-}$ vs. WT mice. These $HSA^{high}Lin^-$ (including $DX5)^-CD4^-CD8^-CD44^-CD25^-$ $TCRβ^+$ cells are not β-selection intermediates, but rather comprise post-selection mature precursor thymocytes of $CD4^-CD8αβ^-$ αβ T cells, found mainly in the gut epithelium (*Pobezinsky et al., 2012*; *Gangadharan et al., 2006*) (and Hilde Cheroutre, personal communication). Their reduction reflects the mature T cell deficiency in $Itpkb^{-/-}$ mice (*Huang et al., 2007*; *Pouillon et al., 2003*; *Wen et al., 2004*). Surface $TCRβ^-$ DN4 cells have all hallmarks of 'true' DN4 cells: They proliferate highly, are metabolically active and efficiently generate DP cells in

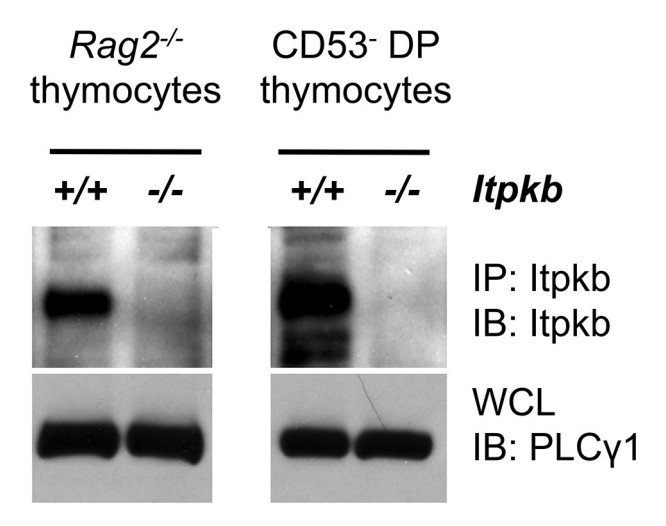

**Figure 1.** Itpkb protein is expressed in DN3 cells from *Itpkb*^+/+ but not *Itpkb*^-/- mice. Shown are immunoblots (IB) of Itpkb immunoprecipitates (IP, top) or whole cell lysates (WCL, bottom) from *Itpkb*^+/+ or *Itpkb*^-/- DN3 cell-enriched *Rag2*^-/- thymocytes (left) or sorted CD53^- DP thymocytes (right), resolved via SDS-PAGE and probed with antibodies against Itpkb (top) or PLCγ1 (bottom, loading control) as in (*Miller et al., 2007*; *Huang et al., 2007*). DOI: 10.7554/eLife.10786.003

vitro (*Petrie et al., 1990*; *Levelt et al., 1993*; *Panigada et al., 2002*; *Kelly et al., 2007*; *Yuan et al., 2011*). So, we used TCRβ^- DN4 cells to further characterize DN4 cell phenotypes. Compared to WT mice, *Itpkb*^-/- mice had significantly less TCRβ^- DN4 and ISP cells, but similar DP cell numbers and an increased (DN3 + DN3-4):TCRβ^- DN4 cell ratio (*Figure 2C*).

pre-TCR expression correlates with upregulated surface CD2, pre-TCR signaling upregulates surface CD5, and surface CD27 upregulation is one of the earliest markers of β-selection (*Taghon et al., 2006*; *Patra et al., 2006*). Compared to WT controls, *Itpkb*^-/- DN3-4 cells and later stages had increased CD2 levels and normal to increased surface levels of CD5, CD27 and CD71 (*Figure 3A*). Also, *Itpkb*^-/- DN3-4 and DN4 cells had elevated surface-levels of the costimulatory chemokine-receptor and Notch-target CXCR4 (*Trampont et al., 2010*; *Xie et al., 2013*). In contrast, *Itpkb*^-/- and WT DN3, DN3-4 and TCRβ^- DN4 cells each expressed comparable surface CD28, CD127 and CD98. Normal to elevated activation markers, and normal DN3 cell and total thymocyte numbers argue against a pre-TCR signaling defect and β-selection block in *Itpkb*^-/- mice but might rather suggest pre-TCR hyper-responsiveness. To further test this, we bred our mice to *Nr4a1/Nur77-GFP* transgenics. Here, Nr4a1/Nur77-GFP expression is a highly sensitive readout for TCR signal intensity (*Moran et al., 2011*). Supporting pre-TCR hyper-responsiveness, DN3 and later stages of thymocyte development expressed more Nr4a1/Nur77-GFP in *Itpkb*^-/- than WT mice (*Figure 3B*).

We next analyzed whether the reduced DN4 and ISP cell numbers in *Itpkb*^-/- mice might reflect reduced proliferation or viability. But similar Ki67-staining and *in vivo* BrdU incorporation suggest comparable steady-state proliferation of all thymocyte subsets between genotypes (*Figure 3C*). Similar AnnexinV staining suggests comparable viability (*Figure 3D*).

## Itpkb-loss cell-intrinsically alters β-selection

To explore if the altered β-selection is caused by thymocyte-intrinsic *Itpkb*-loss, we injected a 1:1 mix of mature T/B cell-depleted *CD45.1* WT and *CD45.2 Itpkb*^-/- BM into lethally irradiated *CD45.1/ CD45.2*-congenic hosts and analyzed reconstituted thymocyte subsets 6–8 weeks later. We distinguished WT vs. *Itpkb*^-/- donor-derived cells by CD45 allelic expression. Compared to WT controls, *Itpkb*^-/- donor-derived thymocytes reproduced the published cell-intrinsic block at the DP stage (*Wen et al., 2004*) and the increased DN3 and reduced DN4 cell proportions, partial loss of TCRβ^- DN4 and ISP cells, and increased (DN3 + DN3-4):TCRβ^- DN4 cell ratio of *Itpkb*^-/- mice (*Figure 4A–C*). Moreover, *Itpkb*^-/- vs. WT donor-derived DN3-4 and DN4 cells overexpressed CD2 and tended

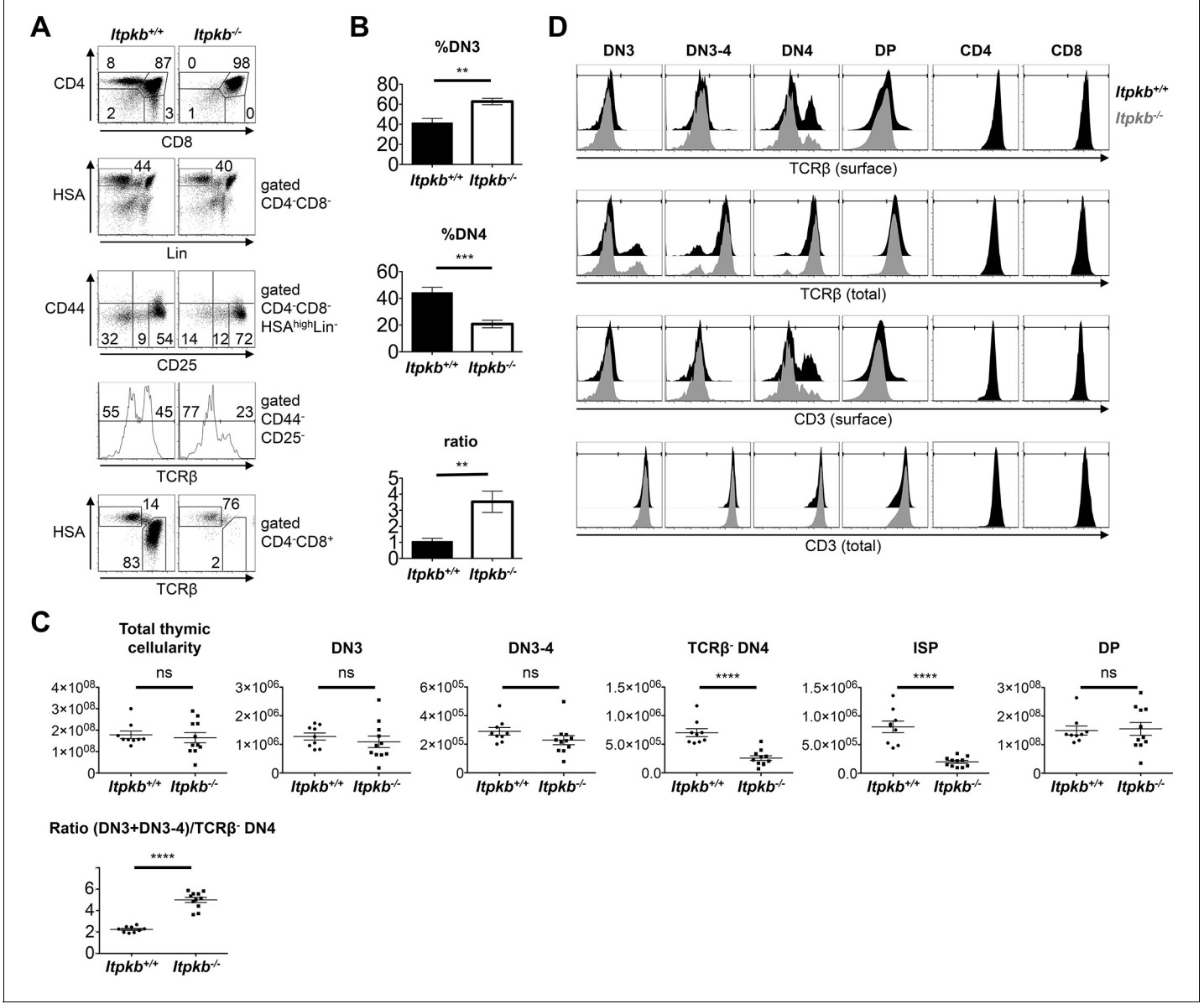

**Figure 2.** Altered *β*-selection in *Itpkb⁻/⁻* mice. (**A**) Flow cytometric profiles of thymocytes from *Itpkb⁺/⁺* and *Itpkb⁻/⁻* littermate mice. Top, CD4 and CD8 expression. Upper center, HSA and mature lineage marker (Lin = CD11b, CD11c, CD19, B220, CD49b, Gr-1, Ter119, TCRγ) expression on CD4⁻CD8⁻ (DN) cells. Lower center, CD44 and CD25 expression on HSA^high Lin⁻ DN cells. The bottom gates denote DN3, transitional DN3-4 and DN4 cells from right to left. Bottom two panels, TCRβ expression on DN4 cells, and HSA and TCRβ expression on CD4⁻CD8⁺ cells with ISP (HSA^high TCRβ^low) and mature CD8 T cell (HSA^low TCRβ^high) gates. Numbers denote % cells per gate. Representative of at least 7 independent experiments. (**B**) Mean ± SEM % DN3 cells, %DN4 cells or %DN3:%DN4 cell ratio in *Itpkb⁺/⁺* or *Itpkb⁻/⁻* 5–7 week old littermates. Statistical significance of genotype differences was analyzed by unpaired two-tailed Student's t-tests (n = 7). (**C**) Total numbers of thymocytes, Lin⁻HSA^high CD44⁻CD25⁺ DN3, CD44⁻CD25^int DN3-4, TCRβ⁻CD44⁻CD25⁻ DN4, CD8α⁺HSA^high TCRβ^low ISP or CD4⁺CD8⁺ DP cells in individual *Itpkb⁺/⁺* or *Itpkb⁻/⁻* mice. Horizontal lines denote means ± SEM. Significance of genotype differences was analyzed as in (**B**). n_WT = 9, n_Itpkb-/- = 11. ns, no significant difference. (**D**) Histograms of surface or total cellular TCRβ or CD3 levels on the indicated thymocyte populations from *Itpkb⁺/⁺* (black) or *Itpkb⁻/⁻* (gray) mice. Representative of ≥3 independent experiments.

to upregulate CXCR4 (*Figure 4D*). Ki67 staining was again similar between genotypes (*Figure 4E*). Thus, *Itpkb⁻/⁻* thymocytes show a cell-intrinsically altered β-selection not rescued by a WT environment and the presence of WT thymocytes.

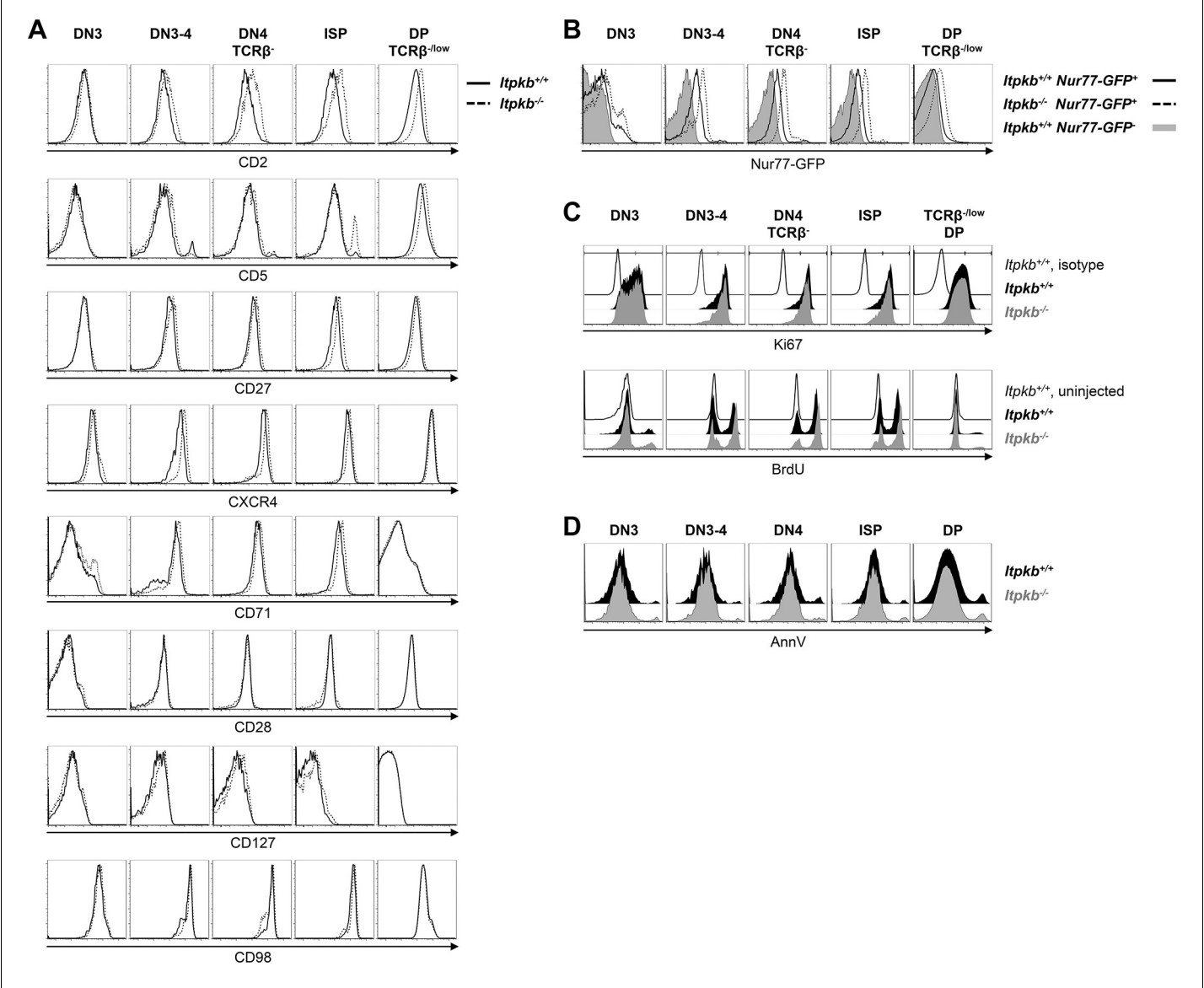

**Figure 3.** Surface marker expression, steady-state proliferation and viability of *Itpkb*[+/+] and *Itpkb*[-/-] thymocytes. (A) Surface levels of the indicated markers for activation or β-selection on thymocyte subpopulations from *Itpkb*[+/+] (solid) or *Itpkb*[-/-] (hatched) mice. (B) Nr4a1/Nur77-GFP expression in *Itpkb*[+/+] or *Itpkb*[-/-] *Nr4a1/Nur77-GFP* transgenic (solid or hatched black, respectively) or non-transgenic (gray) mice. Representative of ≥3 independent experiments. (C) Steady-state proliferative status of the indicated thymocyte subpopulations in *Itpkb*[+/+] (black) or *Itpkb*[-/-] (gray) mice was analyzed by Ki67 stain (top, representative of 3 independent experiments) or BrdU incorporation assay (bottom, representative of 2 independent experiments). Thin open histograms, *Itpkb*[+/+] isotype or BrdU-uninjected, respectively, negative control. TCRβ[low] DP cells were analyzed as they represent the majority of DP cells and *Itpkb*[-/-] mice lack TCRβ[high] DP cells (*Wen et al., 2004*). (D) Steady-state viability of the indicated thymocyte subpopulations in *Itpkb*[+/+] (black) or *Itpkb*[-/-] (gray) mice was analyzed by AnnexinV (AnnV) stain. Representative of 4 independent experiments.

## Itpkb[-/-] DN3 cells develop faster to DP cells

The above data suggest that a developmental block, hypoproliferation or increased death do not cause the loss of DN4 and ISP cells in *Itpkb*[-/-] mice. Another possibility is that these subsets are depleted by accelerated development to DP cells. Indeed, mathematical modeling suggests that a two fold or larger increase in the rate constants for the successive transitions from DN3 to DN4 to ISP to DP cells with unaltered rates for progenitor development to DN3 cells, and for thymocyte turnover due to proliferation and death, can cause similarly reduced DN4 cell and ISP numbers but normal DN3 and DP cell numbers as seen in *Itpkb*[-/-] vs. WT mice (*Figure 5A,B*).

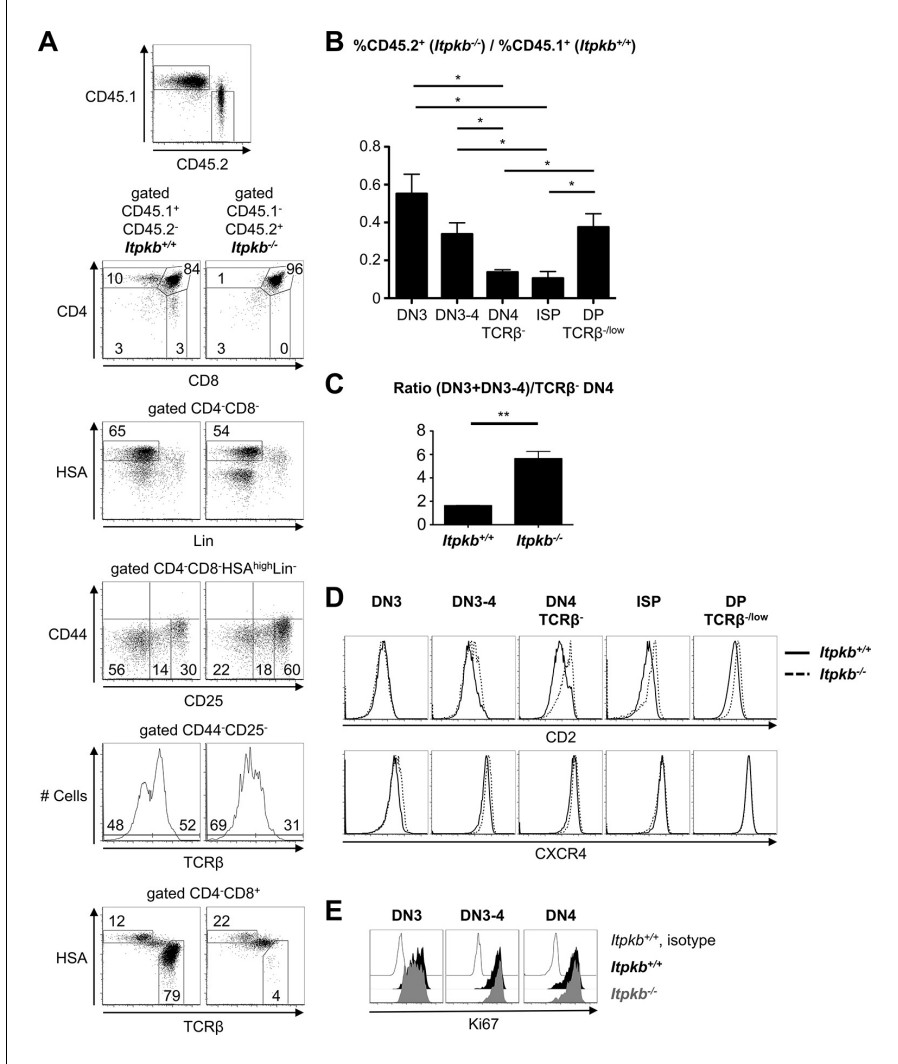

**Figure 4.** Itpkb controls *β*-selection cell-autonomously. B/T cell-depleted BM from *CD45.1 Itpkb*$^{+/+}$ and *CD45.2 Itpkb*$^{-/-}$ mice was mixed at a 1:1 ratio and injected into *CD45.1/CD45.2* lethally irradiated hosts. 7 weeks later, thymocytes were analyzed by FACS. (**A**) Top, thymocyte expression of CD45.1 and CD45.2. The other panels show expression of the indicated markers on CD45.1$^{+}$CD45.2$^{-}$*Itpkb*$^{+/+}$ or CD45.1$^{-}$CD45.2$^{+}$*Itpkb*$^{-/-}$ donor-derived thymocytes, using the gating strategy in *Figure 2A*. Numbers denote % cells per gate. (**B**) Chimerism of the indicated thymocyte subpopulations, expressed as mean ± SEM ratio of CD45.1$^{-}$CD45.2$^{+}$*Itpkb*$^{-/-}$ to CD45.1$^{+}$CD45.2$^{-}$*Itpkb*$^{+/+}$ donor-derived thymocytes. (**C**) Mean ± SEM ratio of total DN3 cell numbers to TCRβ$^{-}$ DN4 cell numbers in *Itpkb*$^{+/+}$ or *Itpkb*$^{-/-}$ donor-derived thymocytes. Significance of the indicated comparisons was analyzed as in *Figure 2* (n = 3). (**D**) CD2 and CXCR4 expression on *Itpkb*$^{+/+}$ (solid) and *Itpkb*$^{-/-}$ (hatched) thymocyte subsets in mixed BM chimeras. Representative of 3 independent hosts. (**E**) Ki67 expression in *Itpkb*$^{+/+}$ (black) and *Itpkb*$^{-/-}$ (gray) DN3, DN3-4 and DN4 cells in mixed BM chimeras. Open histogram, *Itpkb*$^{+/+}$ isotype staining negative control. Representative of 3 independent hosts.

Consistent with faster thymocyte development, *Itpkb*$^{-/-}$ fetal thymi had reduced overall DN and DN4 cell proportions but higher DP cell proportions and total numbers than WT controls on embryonic day 16.5 (E16.5) where DP cells are first detectable in WT mice, and on E17.5 despite 'catching up' WT DP cells (*Figure 5C*, *Figure 5—figure supplement 1*).

To corroborate these findings in an in vivo system that allows one to kinetically study β-selection of a synchronized DN3 cell population, we generated *Itpkb*$^{-/-}$*Rag2*$^{-/-}$mice. *Rag2*-loss causes a developmental arrest at the DN3 stage due to blocked TCRβ expression. Injected anti-CD3 antibodies (α-CD3) can crosslink CD3ε on *Rag*$^{-/-}$ DN3 cells and trigger their differentiation to DP cells

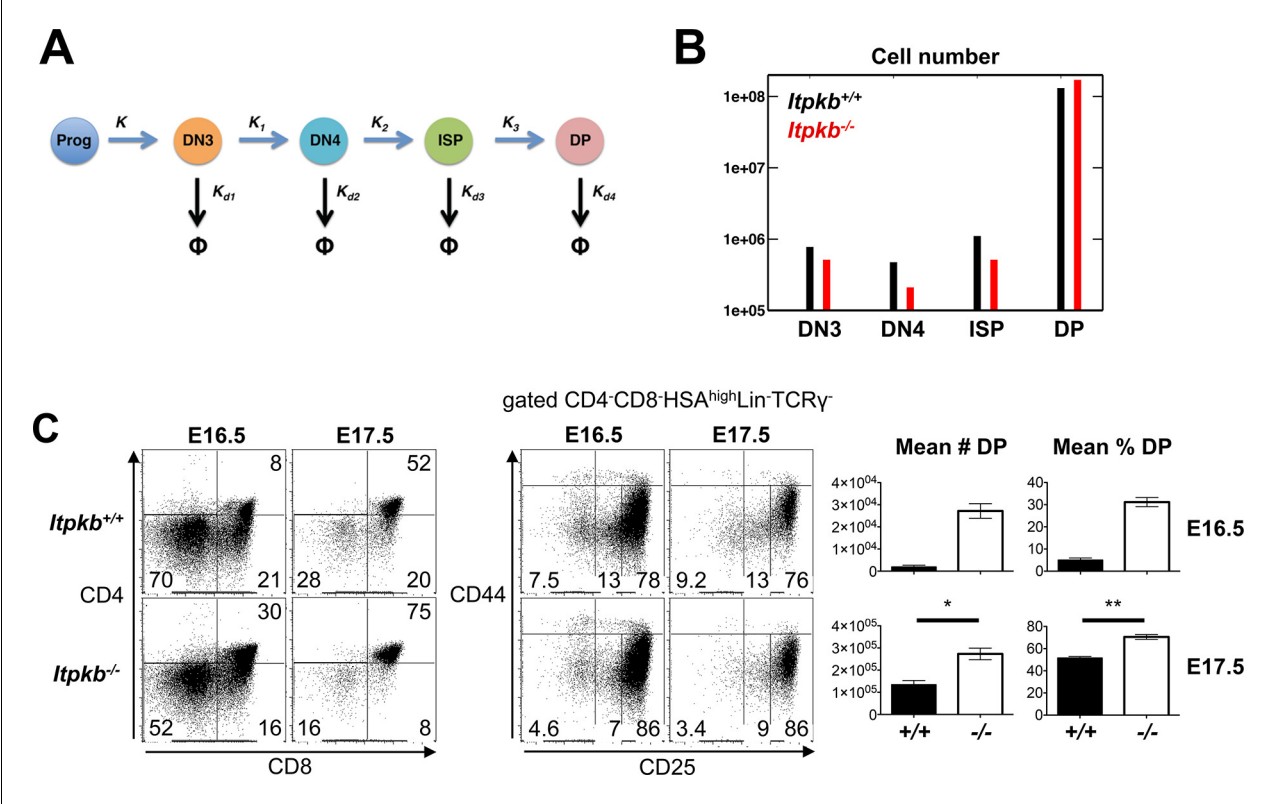

**Figure 5.** Accelerated differentiation of *Itpkb⁻/⁻* DN3 thymocytes. (**A**) In silico analysis of β-selection kinetics in *Itpkb⁻/⁻* and *Itpkb⁺/⁺* mice. Scheme of intrathymic DP cell development from progenitors. The velocities of relevant developmental transitions are characterized by rate constants *K* (identical between genotypes) and $K_1$, $K_2$ and $K_3$ (development from DN3 to DP cells, set to over two-fold higher in *Itpkb⁻/⁻* vs. WT mice, *Table 1*). Rate constants $K_{d1}$-$K_{d4}$ for subset turnover via proliferation and death were considered identical between genotypes. (**B**) Predicted steady-state numbers of the indicated thymocyte populations in *Itpkb⁺/⁺* (black) or *Itpkb⁻/⁻* (red) mice. (**C**) Left, CD4/CD8 expression on embryogenesis day (E) 16.5 or 17.5 fetal thymocytes. Center, CD44/CD25 expression on CD4⁻CD8⁻HSA^high^Lin⁻TCRγ⁻ cells (*Figure 5—figure supplement 1A*). Numbers denote % cells per gate. CD4⁻CD8⁺ fetal thymocytes are ≥92% ISP (*Figure 5—figure supplement 1B*). Right, mean ± SEM DP cell number (#) or % in E16.5 or E17.5 *Itpkb⁺/⁺* or *Itpkb⁻/⁻* thymi. Significance of genotype differences was analyzed as in *Figure 2* ($n_{E16.5}$ = 2, $n_{E17.5}$ = 3). E16.5 data with t-test from another experiment in *Figure 5—figure supplement 1C,D*.

The following figure supplement is available for figure 5:

**Figure supplement 1.** Raw and replicate data related to *Figure 5C*.

(*Campese et al., 2006*). Our mathematical model predicted immediately increasing DN4 cell numbers, and DP cell accumulation at ≥2 days post α-CD3 injection in *Rag2⁻/⁻* mice (*Figure 6A,B*). Simulating over two fold faster DN3-to-DP development in *Rag2⁻/⁻Itpkb⁻/⁻* mice predicted transiently increased DN4 cell accumulation resulting in earlier DP cell accumulation. Experiments confirmed the predictions: Thymocyte development was blocked at the DN3 stage in *Itpkb⁻/⁻Rag2⁻/⁻* and *Rag2⁻/⁻* control mice (*Figure 6C,D*). α-CD3 injection triggered progressive accumulation of DN4, ISP and DP cells 2 and 3 days later in both genotypes (*Figure 6C–F*). However, *Itpkb⁻/⁻Rag2⁻/⁻* mice accumulated larger proportions of DN4 (33% vs. 18%), ISP (3% vs. 1%) and DP cells (20% vs. 1%) than *Rag2⁻/⁻* mice on day 2 when *Rag2⁻/⁻* mice barely had any DP cells. *Itpkb⁻/⁻* DN4 and ISP cell numbers tended to be increased on day 2 but this was not statistically significant (*Figure 6E*). *Itpkb⁻/⁻Rag2⁻/⁻* mice continued to accumulate more DP cells towards an ~four fold excess over *Rag2⁻/⁻* controls on day 3 (*Figure 6F*). This resulted mostly from faster development, as both genotypes showed similar thymocyte subset viability and proliferation post α-CD3 injection (*Figure 6G,H*).

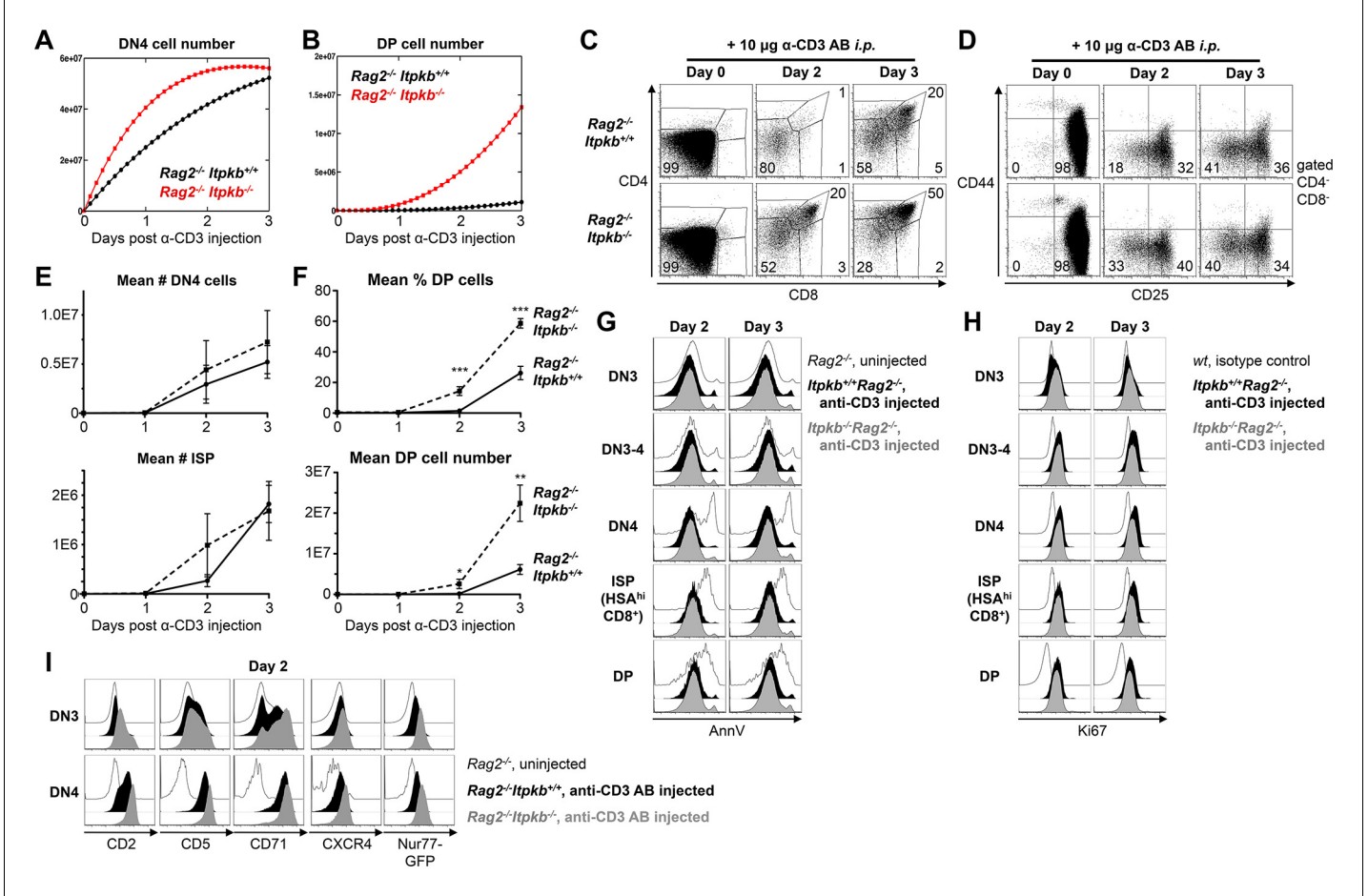

**Figure 6.** Accelerated differentiation of *Itpkb*[-/-] DN3 thymocytes. (**A,B**) Mathematically predicted numbers of DN4 (**A**) and DP cells (**B**) in *Rag2*[-/-]*Itpkb*[+/+] (black) or *Rag2*[-/-]*Itpkb*[-/-] (red) mice on the indicated days post α-CD3 antibody injection. >98% of thymocytes in *Rag2*[-/-] mice are DN3 cells (C,D), and progenitor influx within 3 days is negligible. Thus, we set K = 0 in our model for (A,B). (**C,D**) CD4/CD8 expression on thymocytes (**C**) and CD44/CD25 expression on DN cells (**D**) from *Itpkb*[+/+]*Rag2*[-/-] or *Itpkb*[-/-]*Rag2*[-/-] mice before (day 0), 2 or 3 days post α-CD3 antibody injection. Gates in (**D**) denote DN1, DN2, DN3, DN3-4 and DN4 cells in clock-wise order, numbers % cells in the DN3 and DN4 gates. Representative of 7 independent experiments. (**E**) Measured mean ± SEM DN4 cell (upper panel) and ISP (lower panel) numbers in *Itpkb*[+/+]*Rag2*[-/-] (solid line) or *Itpkb*[-/-]*Rag2*[-/-] (hatched) mice before (day 0) or 1, 2 or 3 days after α-CD3 injection. Significance of the indicated comparisons was analyzed as in *Figure 2*. For DN4 cell numbers, n = 4, 4, 6 or 6 *Rag2*[-/-] and 4, 4, 5 or 5 *Itpkb*[-/-]*Rag2*[-/-] mice, respectively. For ISP numbers, n = 6, 5, 7 or 7 *Rag2*[-/-] and 4, 4, 6 or 5 *Itpkb*[-/-]*Rag2*[-/-] mice, respectively. (**F**) Mean ± SEM DP cell % (upper panel) or number (lower panel) in *Itpkb*[+/+]*Rag2*[-/-] (solid line) or *Itpkb*[-/-]*Rag2*[-/-] (hatched) mice before (day 0) or 1, 2 or 3 days after α-CD3 antibody injection. Significance of genotype differences per day was analyzed as in *Figure 2*. n = 6, 5, 7 or 7 *Rag2*[-/-] and 4, 4, 6 or 5 *Itpkb*[-/-]*Rag2*[-/-] mice, respectively. (**G**) Annexin V (AnnV) staining of DN3, DN3-4, DN4, HSA[hi]CD8[+] ISP or DP cells from uninjected *Rag2*[-/-] (open histograms) or α-CD3 antibody injected *Itpkb*[+/+]*Rag2*[-/-] (black filled histograms) or *Itpkb*[-/-]*Rag2*[-/-] (gray filled histograms) mice two (left) or three (right) days post antibody injection. Representative of 3 independent experiments and 3–4 mice per genotype. Uninjected *Rag2*[-/-] mice contain dying cells in the DN4, CD8-ISP and DP cell gates due to failed β-selection at the DN3 stage. These serve as positive controls for the Annexin V stain. (**H**) Ki67 expression in DN3, DN3-4, DN4, HSA[hi]CD8[+] ISP or DP cells from α-CD3 antibody injected *Itpkb*[+/+]*Rag2*[-/-] (black filled histograms) or *Itpkb*[-/-]*Rag2*[-/-] (gray filled histograms) mice two (left) or three (right) days post antibody injection. Representative of 2 independent experiments and 3 mice per genotype. Open histograms, day 0 WT isotype control. (**I**) CD2, CD5, CD71 and CXCR4 surface-levels on, and transgenic Nr4a1/Nur77-GFP expression in DN3 or DN4 thymocytes from uninjected *Rag2*[-/-] (open histograms) or α-CD3 injected *Itpkb*[+/+]*Rag2*[-/-] (black) or *Itpkb*[-/-]*Rag2*[-/-] (gray) mice 2 days post injection. The <1% CD44[-]CD25[-] negative control cells in uninjected *Rag2*[-/-] mice are non-T cells. Representative of ≥3 independent experiments.

Further supporting accelerated development, *Itpkb*[-/-] sorted DN3 cells generated larger proportions of DP cells than WT DN3 cells after 4-day co-culture on OP9DL1 stroma cells (*Ciofani and Zuniga-Pflucker, 2005*; *Ciofani et al., 2004*) (*Figure 7A,B*, *Figure 9—figure supplement 1*). Finally, *Itpkb*[-/-] E15.5 fetal thymic organ cultures (FTOC) produced more DP cells than WT controls (*Figure 7C,D*).

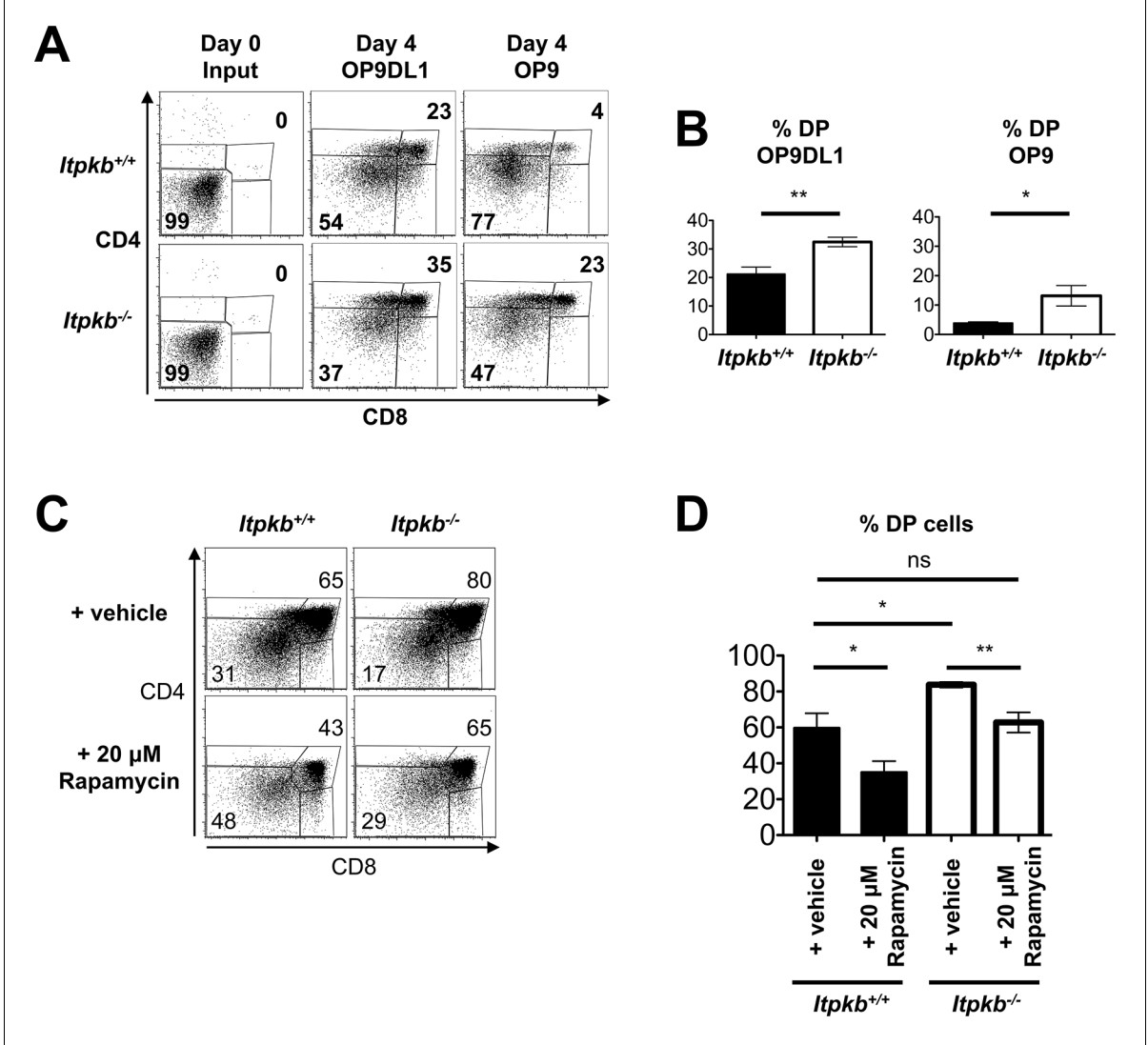

**Figure 7.** *Itpkb*-loss in DN3 cells causes accelerated, Notch-independent development to the DP stage. (A,B) Sorted DN3 cells from 6.5 week old *Itpkb*[+/+] or *Itpkb*[-/-] mice were seeded onto Delta-like 1 Notch ligand-expressing OP9DL1 or Notch ligand-free OP9 stroma cells and analyzed for CD4/CD8 expression 4 days later. (A) Representative FACS data from input (day 0) or day 4 cultures. The numbers indicate % cells in the DP or DN gates, respectively. Representative of 5 independent experiments. (B) Bar-graphs showing mean ± SEM *Itpkb*[+/+] (black bars) or *Itpkb*[-/-] (open bars) % DP cells after 4-day culture on OP9DL1 or OP9 cells, averaged from 4 independent experiments. Significance for genotype differences was analyzed as in *Figure 2* (n = 4). (C,D) Fetal thymic lobes from *Itpkb*[+/+] or *Itpkb*[-/-] embryos harvested on day 15.5 of embryogenesis (E15.5) from the same mother were cultured in the presence of ethanol (vehicle) or 20 μM rapamycin for 4 days, harvested and analyzed. (C) Representative FACS plots of CD4/CD8 expression on total thymocytes. Numbers denote % cells in the respective gate. (D) Bar graph of mean ± SEM % DP cells for each condition and genotype from 3 independent experiments. Significance of the indicated comparisons was analyzed as in *Figure 2* (n = 5).

## Increased pre-TCR signaling via PI3K/Akt/mTOR in *Itpkb*[-/-] DN3 cells

Activation marker upregulation and faster DN-to-DP cell development suggest increased pre-TCR signaling in *Itpkb*[-/-] mice. To confirm this, we analyzed the α-CD3 induced upregulation of pre-TCR activation markers in the *Rag2*[-/-] mouse system. Compared to uninjected *Rag2*[-/-] controls, α-CD3 injection upregulated surface CD2, CD5, CD71 and CXCR4 on DN3 and, more pronounced, on DN4 cells in *Rag2*[-/-] and *Itpkb*[-/-] *Rag2*[-/-] mice 2 days later (*Figure 6I*). All markers reached higher surface-levels on *Itpkb*[-/-]*Rag2*[-/-] than *Rag2*[-/-] DN3 and DN4 cells. Thus, *Itpkb*[-/-] DN3/DN4 cells are hyperresponsive to CD3-crosslinking. To determine if signaling is increased, we injected α-CD3 into *Nr4a1/*

*Nur77-GFP* transgenic *Rag2*⁻/⁻ and *Itpkb*⁻/⁻*Rag2*⁻/⁻ mice. Higher Nr4a1/Nur77-GFP induction in *Itpkb*⁻/⁻ *Rag2*⁻/⁻ than *Rag2*⁻/⁻ mice confirmed increased CD3 signaling (*Figure 6I*).

Conditionally *Pten*⁻/⁻ DN cells show constitutively active Akt and accelerated development to DP cells (*Hagenbeek et al., 2004*; *Kelly et al., 2007*; *Shiroki et al., 2007*; *Wong et al., 2012*). Itpkb can limit receptor-mediated Akt activation through $IP_4$/$PIP_3$ antagonism (*Sauer and Cooke, 2010*). To test if this mechanism restricts pre-TCR responses, we compared signaling via PI3K, Akt and downstream mTOR in *Itpkb*⁻/⁻ and WT thymocyte subsets. Compared to negative controls and positive controls treated with the phosphatase-inhibitor Calyculin A (*Pozuelo-Rubio et al., 2010*), WT and *Itpkb*⁻/⁻ TCRβ⁻ pre-selection DN3 cells had significant but similar basal amounts of phosphorylated active Akt, mTOR and downstream S6 protein (*Figure 8A*). TCRβ⁺ DN3 cells undergoing β-selection upregulated phospho-Akt, -mTOR, -S6 and control -Erk, indicating pre-TCR mediated activation. Importantly, and contrasting with a WT-like total Akt protein content in *Itpkb*⁻/⁻ cells, phospho-Akt, -mTOR and -S6 but not -Erk levels were higher in *Itpkb*⁻/⁻ than WT TCRβ⁺ DN3 and DN3-4 cells (*Figure 8A*). This genotype difference disappeared as signaling was downregulated in later developmental stages or in mature TCRβ⁺ 'DN4-phenotype' gut T cell precursors.

Akt/mTOR-activation by the pre-TCR and Notch promotes DN3/DN4 cell metabolism in part by increasing nutrient import through upregulation or activation of the glucose-transporter Glut1, the L-amino acid transporter CD98 and the transferrin receptor CD71 on the cell surface. This results in an increased cell size (*Ciofani and Zuniga-Pflucker, 2005*; *Kelly et al., 2007*; *Fayard et al., 2010*; *Juntilla and Koretzky, 2008*; *Janas et al., 2010*). Metabolic activation appeared increased in *Itpkb*⁻/⁻ TCRβ⁺ DN3 and DN3-4 cells, because they showed Glut-1 hyperinduction and increased large cell proportions over WT controls (*Figure 8A,B*). Moreover, α-CD3 injection hyperinduced surface CD71 on DN3 and DN4 cells in *Itpkb*⁻/⁻*Rag2*⁻/⁻ versus *Rag2*⁻/⁻ mice (*Figure 6I*). Taken together, these data suggest that Itpkb limits PI3K/Akt/mTOR signaling in, and metabolic activation of surface TCRβ⁺ DN3 and DN3-4 cells.

## Itpkb restricts pre-TCR signaling to delay β-selection and render it Notch-dependent

To determine if the Akt/mTOR and metabolic hyperactivation of *Itpkb*⁻/⁻ pre-TCR⁺ DN3 cells causes their accelerated development to DP cells, we next studied if treatment with inhibitors of Akt (Akt-inhibitor VIII), mTOR (rapamycin) or glucose-metabolism (2-deoxy-D-glucose, 2DG) could reverse the increased DP cell generation from equal numbers of sorted *Itpkb*⁻/⁻ versus *Itpkb*⁺/⁺ DN3 cells on OP9DL1 stroma. Strikingly, all three treatments yielded *Itpkb*⁻/⁻ DP cell proportions similar to those of untreated or carrier-treated WT controls after 4-day co-culture (*Figure 9A*, *Figure 9—figure supplement 1A,C,E*). As expected, the treatments also reduced WT DP cell generation below untreated controls. Their reduced efficacy towards *Itpkb*⁻/⁻ DN cells is expected, as these had increased amounts of the respective active inhibitor-targets (*Figure 8A*). Similarly complete rapamycin reversal of the accelerated DN-to-DP development in *Itpkb*⁻/⁻ versus *Itpkb*⁺/⁺ FTOC confirmed these important findings in a less reductionist system (*Figure 7C,D*). These data suggest that the hyperactive Akt, mTOR and glucose metabolism of *Itpkb*⁻/⁻ DN3 cells contribute to their accelerated DN-to-DP development.

Constitutive Akt activity promotes glucose metabolism and allows DN3-to-DP cell maturation without Notch signaling (*Ciofani and Zuniga-Pflucker, 2005*; *Lee et al., 2012*; *Fayard et al., 2010*; *Juntilla and Koretzky, 2008*). To test if the Akt/mTOR hyperactivity in *Itpkb*⁻/⁻ TCRβ⁺ DN3 cells has the same effect, we co-cultured sorted *Itpkb*⁻/⁻ or WT DN3 cells with OP9 stroma lacking Notch ligands (*Figures 7A,B*, *9B*, *Figure 9—figure supplement 1B,D,F*). As previously described (*Ciofani and Zuniga-Pflucker, 2005*; *Xiong et al., 2011*), WT DN3 cells much less efficiently generated DP cells without than with Notch (*Figures 7A,B*, *9A,B*). Inhibition of Akt, mTOR or Glucose-metabolism further reduced WT DP cell production consistent with known Akt, mTOR and glycolysis requirements for DN thymocyte survival and differentiation (*Lee et al., 2012*; *Fayard et al., 2010*; *Juntilla and Koretzky, 2008*). In striking contrast, *Itpkb*⁻/⁻ DN3 cells efficiently generated DP cells on OP9 stroma (*Figures 7A,B*, *9B*, *Figure 9—figure supplement 1B,D,F*). Inhibition of Akt, mTOR or glycolysis strongly reduced DP cell output. Thus, *Itpkb*-loss renders the DN3-to-DP transition Notch-independent in an at least partially Akt, mTOR and glycolysis-dependent manner.

Notch signaling depends on its cleavage by cellular γ-secretases (*Wong et al., 2004*). To corroborate our findings *in vivo*, we thus analyzed DN3 cell maturation in *Itpkb*⁻/⁻*Rag2*⁻/⁻ versus *Rag2*⁻/⁻ mice

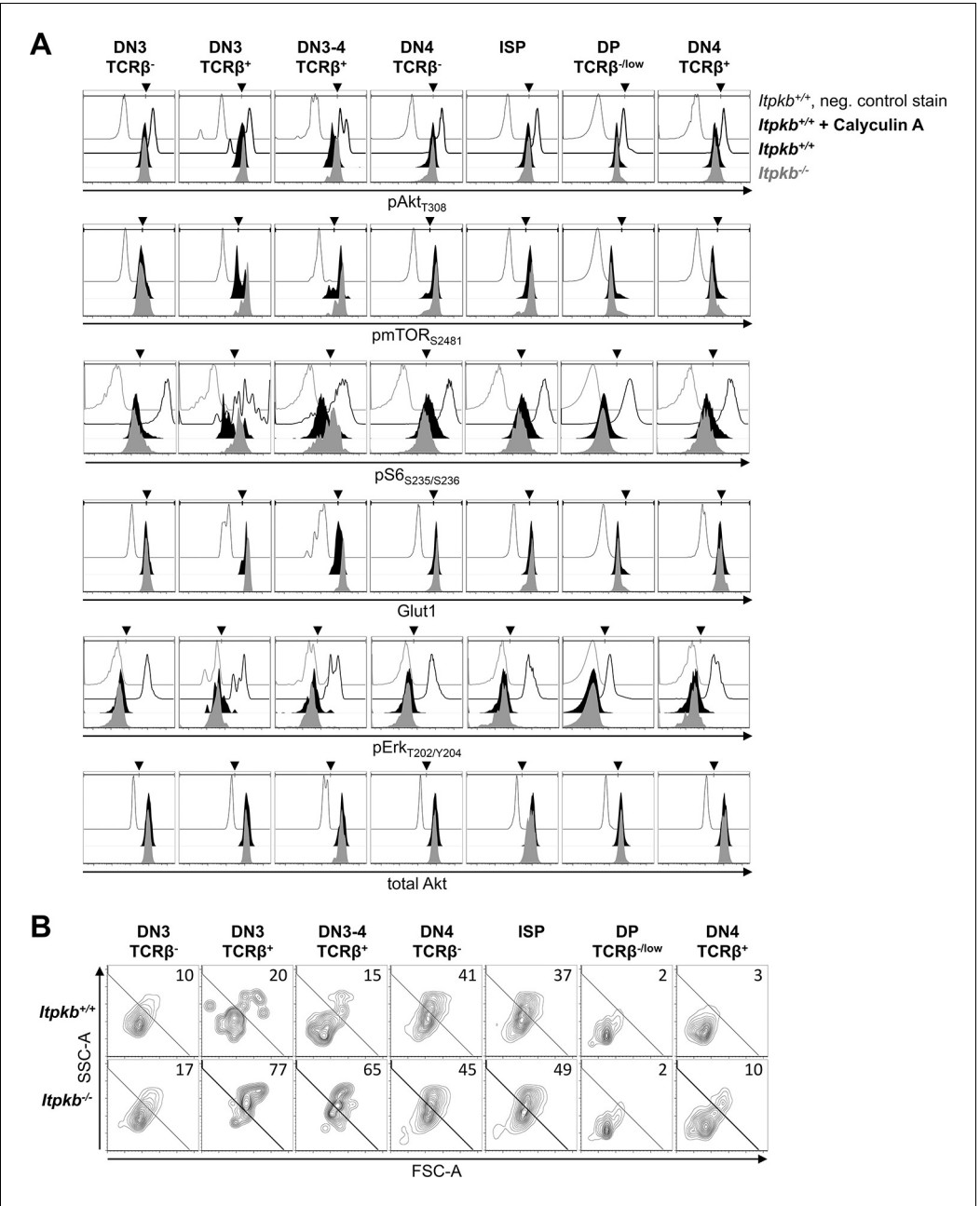

**Figure 8.** Increased pre-TCR signaling via PI3K/Akt/mTOR in *Itpkb*$^{-/-}$DN3 cells. (**A,B**) We analyzed (**A**) cellular content of T$_{308}$-phosphorylated active Akt (pAkt$_{T308}$), S$_{2481}$-phosphorylated mTOR (pmTOR$_{S2481}$), S$_{235}$/S$_{236}$-phosphorylated ribosomal protein S6 (pS6$_{S235/S236}$), Glut1 protein, T$_{202}$/Y$_{204}$-phosphorylated Erk (pErk$_{T202/Y204}$) and Akt protein, and (**B**) cell size via side/forward-scatter analysis (SSC-A/FSC-A) in the indicated thymocyte populations of *Itpkb*$^{+/+}$ (black histograms) or *Itpkb*$^{-/-}$ (gray histograms) mice by FACS. Thin open histograms, *Itpkb*$^{+/+}$ isotype or second antibody stained negative controls. Bold open histograms, Calyculin A-treated positive controls. Arrowheads show gate positions. In (**B**), numbers indicate % cells per large cell gate. Representative of at least 2 (pS6$_{S235/S236}$), 3 (total Akt) or 8 (else) independent experiments.

treated *p.o.* with vehicle or the γ-secretase inhibitor LY-411,575 for two days post α-CD3 injection. *P.o.* administered 10 mg/kg LY-411,575 potently inhibited γ-secretase function in mice and impaired DN thymocyte maturation into αβ T cells with a particularly profound loss of DP cells (*Wong et al., 2004*). We found that LY-411,575 strongly impaired the α-CD3 induced DN cell development into

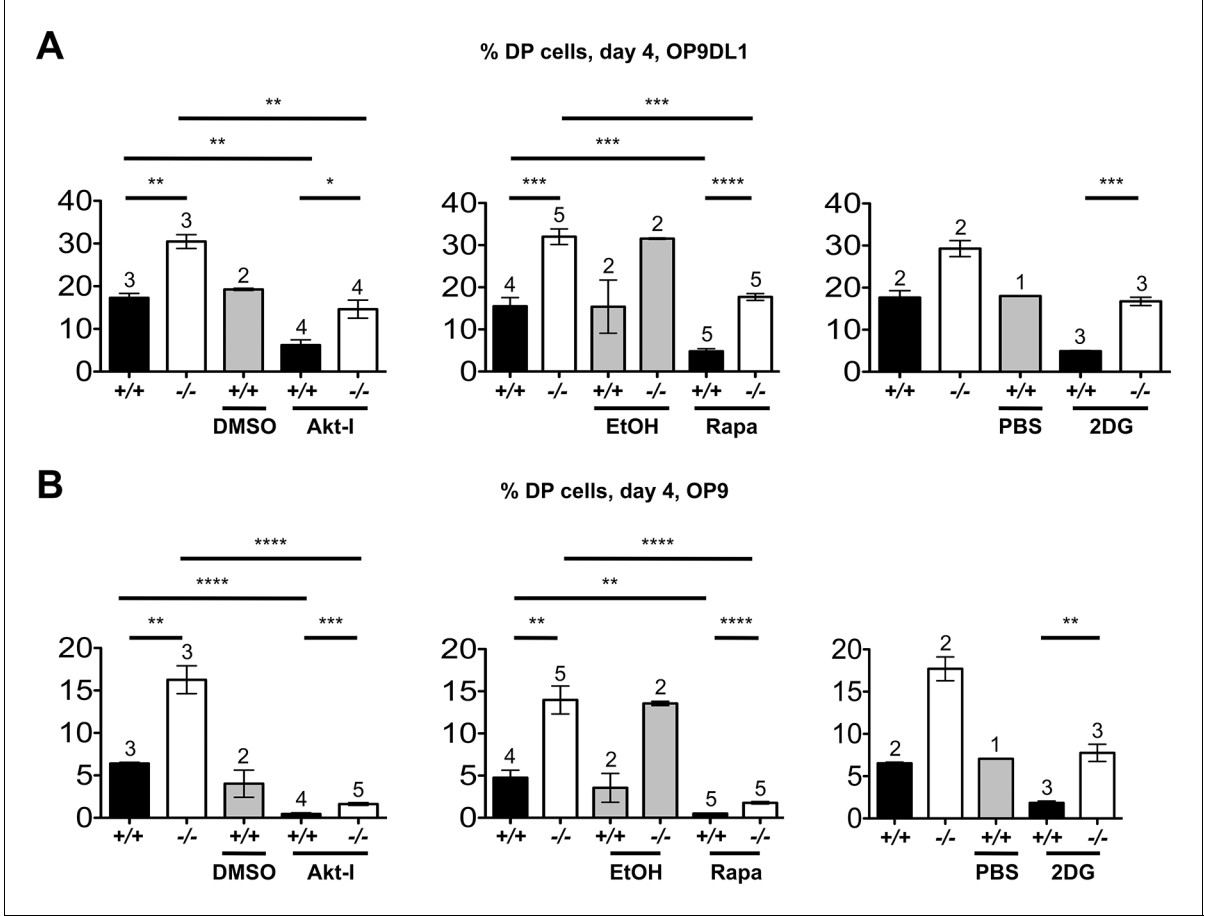

**Figure 9.** Itpkb renders β-selection Notch-dependent. (**A,B**) Addition of inhibitors of Akt, mTOR or glucose metabolism reverses the accelerated development of *Itpkb⁻/⁻* DN3 cells and re-establishes Notch-dependence. Shown are mean ± SEM *Itpkb⁺/⁺* (solid black bars) or *Itpkb⁻/⁻* (pen bars) % DP cells after 4-day culture on OP9DL1 (**A**) or OP9 (**B**) cells without or with addition of carrier (solid gray bars; DMSO, ethanol or PBS, respectively), 500 nM Akt-inhibitor VIII in DMSO (Akt-I, added once on day 0), 4 µM rapamycin in ethanol (Rapa, added once on day 0) or 500 µM 2-deoxy-D-glucose in PBS (2DG, added once daily), averaged from 3 (Akt-I), 4 (rapamycin) or 2 (2DG) independent experiments. Significance of the indicated comparisons was analyzed as in *Figure 2*. Replicate numbers are indicated above each bar. Representative FACS-data in *Figure 9—figure supplement 1*.

The following figure supplement is available for figure 9:

**Figure supplement 1.** Raw FACS data from one representative experiment included in the averaged data in *Figure 9*.

ISP and DP cells in *Rag2⁻/⁻* but not *Itpkb⁻/⁻Rag2⁻/⁻* mice (*Figure 10*). Hence, *Itpkb*-loss reduces the Notch-dependence of DN thymocyte development to DP cells both *in vitro* and *in vivo*.

## Discussion

Here, we identify Itpkb as a novel pre-TCR effector which restricts the kinetics of β-selection and establishes its Notch-dependence. *Itpkb⁻/⁻* mice show a cell-autonomously accelerated and Notch independent, but pre-TCR dependent DN3-to-DP cell differentiation associated with DN3 cell pre-TCR hyperresponsiveness, Akt/mTOR hyperactivation and evidence for metabolic hyperactivity. Pharmacologic inhibition of Akt, mTOR or glucose metabolism restored WT kinetics and Notch-dependence of *Itpkb⁻/⁻* DN3-to-DP cell development.

In thymocytes, TCR engagement activates Itpkb to produce $IP_4$. *Itpkb⁻/⁻* thymocytes had strongly reduced $IP_3$ 3-kinase activity and $IP_4$ levels, but normal $IP_3$ levels and $Ca^{2+}$ mobilization (*Huang et al., 2007*; *Pouillon et al., 2003*; *Wen et al., 2004*). $IP_4$ competitively limits $PIP_3$-binding to the Akt PH domain and Akt activation in NK cells, myeloid cells and HSC (*Jia et al., 2008*;

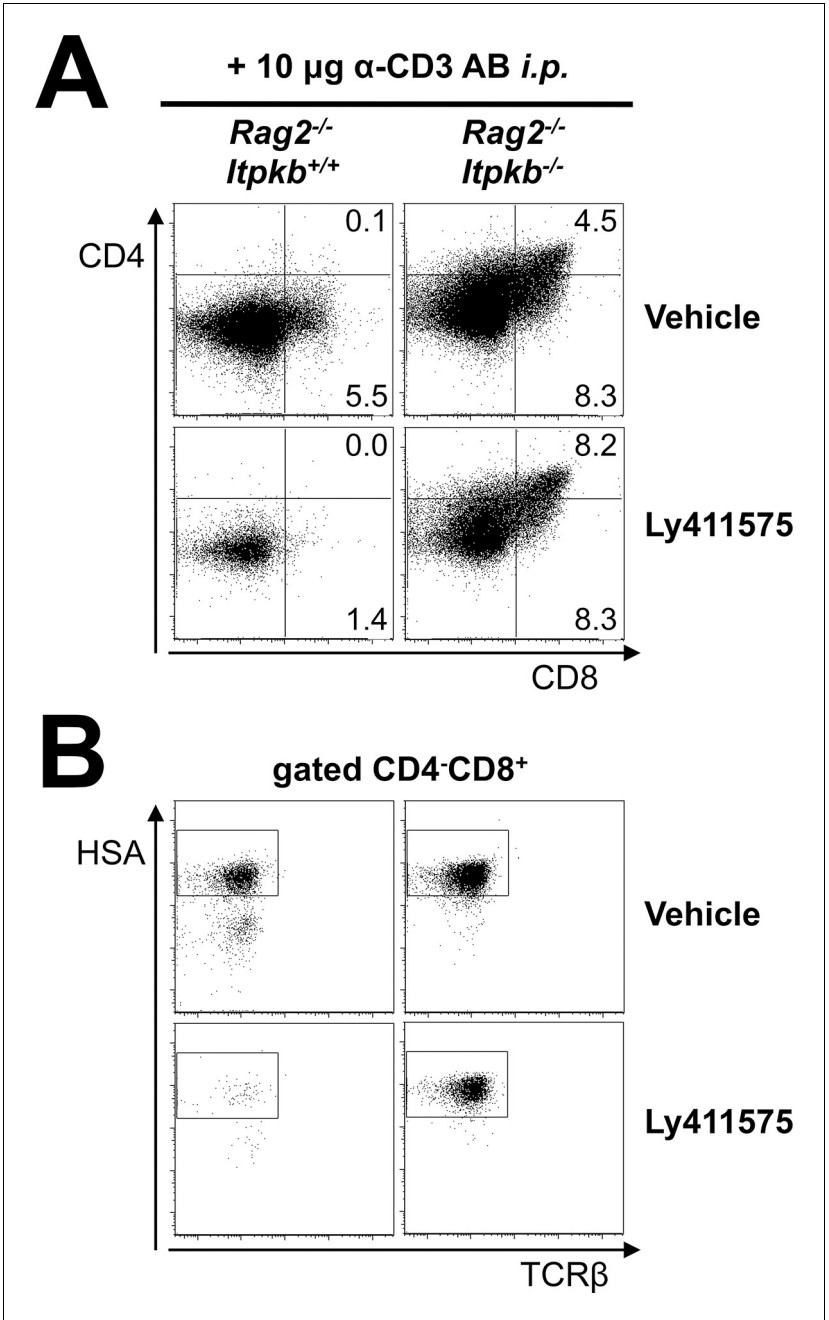

**Figure 10.** *Itpkb*-loss reduces the Notch-dependence of DN thymocyte development to DP cells in vivo. Shown are (**A**) CD4/CD8 expression on total thymocytes and (**B**) HSA/TCRβ expression on CD4⁻CD8⁺ thymocytes from *Rag2⁻/⁻* and *Rag2⁻/⁻Itpkb⁻/⁻* mice two days post α-CD3 antibody injection. Starting 3–4 hr before α-CD3 injection, the mice were treated once daily with orally administered γ-secretase inhibitor LY-411,575 or vehicle (*Wong et al., 2004*). Numbers indicate % cells per respective gate. The gates in (**B**) denote CD8⁺HSA^high ISP (*Petrie and Zuniga-Pflucker, 2007*; *Xiong et al., 2011*). Representative of two independent experiments (n = 3).

*2007*; *Sauer et al., 2013*; *Siegemund et al., 2015*). Thus, we propose that pre-TCR induced $IP_4$/$PIP_3$ antagonism governs β-selection by restricting PI3K/Akt/mTOR signaling and metabolic activation. We derive a model where Itpkb controls pre-TCR/Notch crosstalk through combined restriction of pre-TCR induced and Notch induced PI3K signaling via Akt (*Figure 11*). Itpkb enforced coincidence detection of pre-TCR surface expression and Notch-engagement ensures that Akt is only

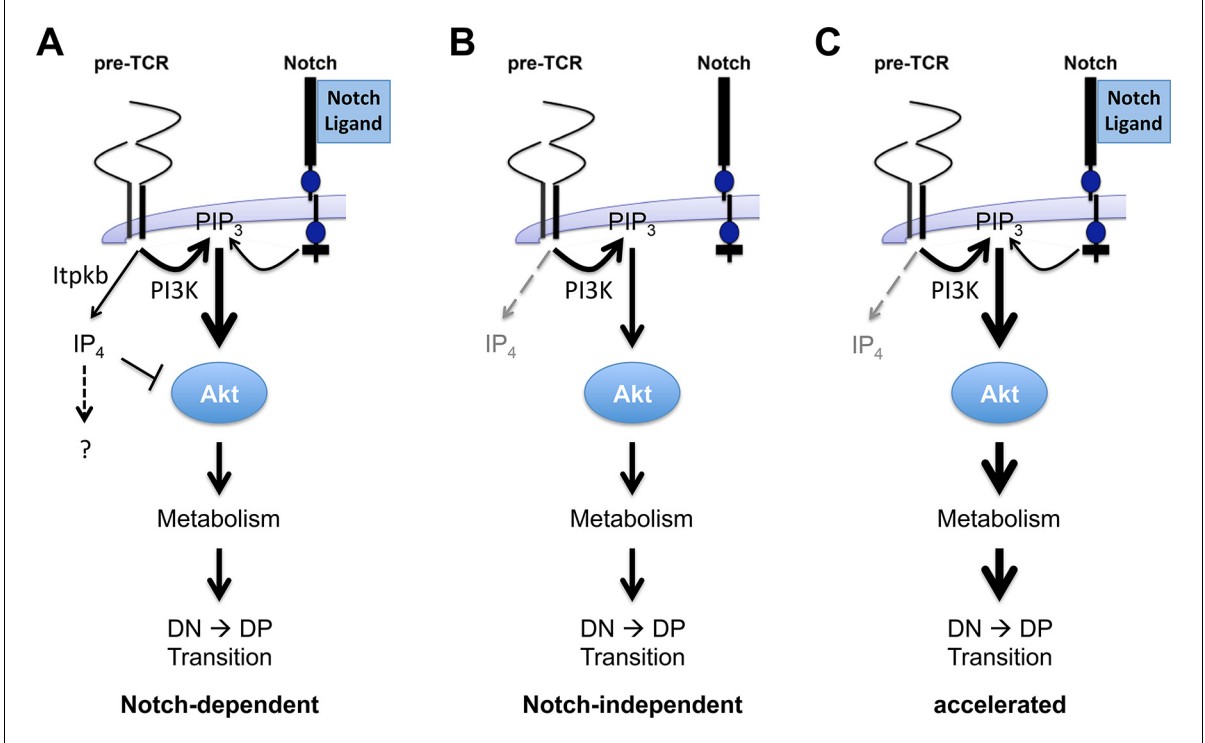

**Figure 11.** Antagonistic signaling by PI3K and Itpkb controls the kinetics and Notch-dependence of $\beta$-selection. (**A**) We propose a model in which pre-TCR and Notch signaling both activate PI3K to produce $PIP_3$ in DN3/DN3-4 cells. $PIP_3$ then recruits and activates Akt to increase glucose metabolism via the Akt/mTOR pathway. This is required for DN3-to-DP cell differentiation. However, pre-TCR signaling also activates Itpkb to produce $IP_4$, which competes with $PIP_3$ for Akt PH domain binding and limits Akt recruitment, Akt and mTOR activation in pre-TCR expressing DN3/DN3-4 cells. $IP_4$ may have additional effectors, indicated by the question mark. By limiting downstream glucose metabolism, this "$IP_4$ brake" delays the kinetics of $\beta$-selection and renders this process dependent on Notch costimulation. (**B**) Without Itpkb, $IP_4$ no more dampens Akt activation and pre-TCR signaling alone sufficiently activates Akt/mTOR signaling to trigger DP cell development in the absence of Notch engagement. (**C**) In the presence of Notch-signals, Akt is now hyperactivated and causes an accelerated DN3-to-DP cell differentiation.

activated to the degree needed for $\beta$-selection and only in an appropriate context, pre-TCR$^+$ DN3 cells interacting with Notch-ligand expressing subcapsular stromal cells (*Petrie and Zuniga-Pflucker, 2007*). This prevents premature differentiation. Similarly accelerated DN-to-DP cell development of *Itpkb$^{-/-}$* and *Pten$^{-/-}$* thymocytes (*Hagenbeek et al., 2004*; *Shiroki et al., 2007*) and enhanced DP cell production from DN3 cells expressing a dominant-active mutant version of the class I PI3K regulatory subunit p85$\alpha$/Pik3r1 (*p65$^{PI3K}$* transgenic mice) (*Rodriguez-Borlado et al., 2003*) or dominant-active, myristoylated Akt1 (*myr-Akt* transgenic mice) (*Lee et al., 2012*) highlight the importance of restricting PI3K signaling via Akt for proper $\beta$-selection kinetics, even though the specific purpose of delaying DP cell maturation remains unknown.

The increased DP cell production from *Itpkb$^{-/-}$* versus WT DN3 thymocytes in several different *in vivo* and *in vitro* models without concomitantly increased proliferation or viability suggests accelerated developmental kinetics, consistent with our mathematical simulations. We propose that this is caused by pre-TCR hyperresponsiveness based on the Akt/mTORC1 hyperactivation, increased Nr4a1/Nur77-GFP expression and hyperinduction of activation markers in *Itpkb$^{-/-}$* versus WT DN3 and later stage thymocytes, and on the phenotype reversal by Akt/mTORC1 and metabolic inhibitors. Importantly, the Nr4a1/Nur77-GFP and activation marker hyperinduction in *Itpkb$^{-/-}$* DN3 cells indicate increased transcriptional responses. This might accelerate development by inducing required amounts of cell fate determinants earlier in *Itpkb$^{-/-}$* DN3 cells than in WT cells. Alternatively or in addition, *Itpkb*-loss might increase the number of DN3 cells responding to pre-TCR signals or developmental cues present even in the reductionist OP9-DL1 cell co-culture system. Our present data do not allow us to discern the relative contributions of accelerated kinetics of cellular signaling

events versus increased proportions of responding cells due to lowered pre-TCR signaling thresholds or enhanced sensitivity to developmental cues upon *Itpkb*-loss. Distinguishing between these possibilities will require future detailed studies of the effects of *Itpkb*-loss on the sizes (amplitudes and proportions of cells responding), kinetics (rate constants) and shapes (analog, digital) of pre-TCR signaling events, transcriptional and functional responses in populations of individually analyzed DN3 cells, combined with mathematical simulations.

pre-TCR and Notch signaling both promote DN3 cell proliferation, survival and differentiation in part by activating PI3K/Akt/mTOR (*Janas et al., 2010*; *Ciofani et al., 2004*; *Ciofani and Zuniga-Pflucker, 2005*; *Taghon et al., 2006*; *Lee et al., 2012*; *Kelly et al., 2007*). The dependence of β-selection on combined pre-TCR and Notch signaling relies on dampened PI3K/Akt signaling (*Juntilla and Koretzky, 2008*; *Fayard et al., 2010*): Conditionally *Pten*$^{-/-}$ DN cells have constitutively active Akt and generate DP cells without pre-TCR or Notch-signaling (*Hagenbeek et al., 2004*; *Kelly et al., 2007*; *Shiroki et al., 2007*; *Wong et al., 2012*). And constitutive Akt activity can substitute for Notch or mTORC2 to promote DN cell glycolysis, survival and differentiation. It allows DN3-to-DP cell development without pre-TCR or Notch-signaling, but not without both (*Mao et al., 2007*; *Ciofani and Zuniga-Pflucker, 2005*; *Kelly et al., 2007*; *Lee et al., 2012*). Although many details of how Notch and PI3K intersect remain unclear, Notch may promote β-selection in part by inducing HES1 to repress Pten, and c-Myc to promote proliferation (*Wong et al., 2012*).

Contrasting with *Pten*-loss, Notch or Akt hyperactivity, *Itpkb*-loss accelerates DN3 cell differentiation without significant effects on proliferation and viability, and overcomes the Notch dependence but not the pre-TCR dependence of β-selection. This is evidenced by the lack of accumulating intracellular TCRβ$^-$ DP cells in *Itpkb*$^{-/-}$ mice, and of post-DN3 cells in *Itpkb*$^{-/-}$*Rag2*$^{-/-}$ mice (*Figures 2D*, *6C*, *D*). We speculate that this reflects the need for TCR signals to activate Itpkb and produce IP$_4$ (*Chamberlain et al., 2005*; *Wen et al., 2004*; *Pouillon et al., 2003*). By abrogating pre-TCR induced IP$_4$-inhibition of pre-TCR and Notch induced Akt/mTOR signaling, *Itpkb*-loss mimics the effects of *Pten*-loss or dominant active *Akt1* expression. This overcomes Notch-requirements and accelerates differentiation but not proliferation, because Notch-induction of c-Myc is PI3K-independent (*Wong et al., 2012*). The surprising lack of increased DN3/DN4 cell viability in *Itpkb*$^{-/-}$ mice might reflect differing degrees of Akt/mTOR hyperactivation in *Pten*$^{-/-}$, dominant active *Akt1*-expressing and *Itpkb*$^{-/-}$ DN3/DN4 cells, consistent with unaltered development of *Inpp5d/SHIP1*$^{-/-}$ thymocytes (*Kashiwada et al., 2006*). Altogether, the largely restored developmental kinetics and Notch-dependence of *Itpkb*$^{-/-}$ DN3 cells by treatment with Akt, mTORC1 or metabolic inhibitors support contributing roles for the Akt/mTOR-hyperactivity. Future studies with sub-optimal Akt/mTOR inhibitor concentrations not affecting WT thymocytes but still reversing the *Itpkb*$^{-/-}$ phenotype, with complex genetic models and with inhibitors of β-selection effectors unaffected by Itpkb will be needed to more conclusively distinguish between specific causative roles for the Akt/mTORC1 and metabolic hyperactivity and mere remaining sensitivity of *Itpkb*$^{-/-}$ DN thymocytes to inhibition of this particular pathway. Such studies can also address whether additional mechanisms contribute to the β-selection phenotype of *Itpkb*$^{-/-}$ mice.

Contrasting with dominant active Akt1 expression or loss of Pten, which has high constitutive PIP$_3$-phosphatase activity (*Leslie and Foti, 2011*), Itpkb-loss cannot replace pre-TCR signals because Itpkb is inactive without them, so its loss has no further effect. *Itpkb*-loss might also reduce less essential positive Itpkb roles in pre-TCR signaling, such as augmenting PLCγ1/Erk activation by Itk (*Huang et al., 2007*). Indeed, TCRβ$^+$ DN3 cells from *Itpkb*$^{-/-}$ vs. WT mice tended to have mildly reduced Erk activity (*Figure 8A*). Erk signaling is required for DN cell proliferation and differentiation (*Kortum et al., 2013*). The mild defects in *Itpkb*$^{-/-}$ mice are consistent with the only minor role of Itk in β-selection (*Lucas et al., 2007*) and the unaltered DN cell proliferation.

Hyper-upregulation of Glut1, CD71 and cell-size in *Itpkb*$^{-/-}$ TCRβ$^+$ DN3 cells and reversal of their accelerated, Notch-independent differentiation by the glycolytic inhibitor 2DG suggest that Itpkb controls β-selection by ultimately restricting DN3 cell metabolic activation. Similar Akt-inhibitor and rapamycin effects indicate a causative role for Akt/mTOR hyperactivation. Akt promotes metabolism by increasing Glut1 expression and activity, regulating enzymes in glucose and lipid metabolism and promoting mTOR-dependent protein translation (*Juntilla et al., 2007*). In DN cells, Pdk1/Akt/mTORC1 also upregulate surface CD71 and CD98 downstream of pre-TCR and Notch (*Kelly et al., 2007*; *Fayard et al., 2010*). Thus, upregulated iron uptake, glucose and amino acid metabolism and

protein biosynthesis might all contribute to the accelerated, Notch-independent development of *Itpkb*$^{-/-}$ DN3 cells.

Excessive Notch signaling causes thymocyte transformation and T-ALL. This is augmented by pre-TCR signals (*Campese et al., 2006*; *Fayard et al., 2010*). Excessive Akt activity in thymocytes due to PI3K hyperactivity, Pten inactivation or dominant-active *Akt1* expression causes leukemia/lymphoma (*Aifantis et al., 2006*; *Fayard et al., 2010*). The intermediate β-selection phenotype of *Itpkb*-loss between those of *Pten*-loss (*Hagenbeek et al., 2004*; *Kelly et al., 2007*; *Shiroki et al., 2007*; *Wong et al., 2012*) and *Inpp5d/SHIP1*-loss (*Kashiwada et al., 2006*) raises the possibility that IP$_3$ 3-kinases could have tumor suppressor functions by limiting Akt signaling. But we have not seen thymocyte neoplasia or accumulation of intracellular TCRβ$^-$ DP cells in *Itpkb*$^{-/-}$ mice. One possible explanation consistent with low residual IP$_3$ 3-kinase activity and IP$_4$-production in *Itpkb*$^{-/-}$ thymocytes (*Wen et al., 2004*; *Pouillon et al., 2003*) is partial *Itpkb* redundancy with other IP$_3$ 3-kinases. Moreover, their premature lethality due to infections (*Pouillon et al., 2003*) and anemia (*Siegemund et al., 2015*) limits aging studies with *Itpkb*$^{-/-}$ mice. It will be important to re-assess in a germ-free vivarium whether conditional *Itpkb*-disruption in thymocytes causes T-ALL as the mice age, or on a sensitized *Trp53*$^{-/-}$ background as seen for p65$^{PI3K}$ transgenics (*Borlado et al., 2000*). Then again, IP$_4$-loss might simply not augment PIP$_3$ cellular activity sufficiently to transform thymocytes, reminiscent of *Inpp5d/SHIP1*$^{-/-}$-loss (*Kashiwada et al., 2006*). Also, the potential reduction of required Akt-unrelated IP$_4$ functions such as promoting Itk/Erk signaling (*Huang et al., 2007*) might prevent thymocyte transformation in *Itpkb*$^{-/-}$ mice. Clearly, more studies are needed to assess the tumor suppressor potential of Itpks.

By unveiling Itpkb antagonism with PI3K/Akt/mTOR signaling as a key determinant of the kinetics and Notch-dependence of thymocyte β-selection, our findings expand our limited knowledge about physiological IP$_3$ 3-kinase functions (*Sauer and Cooke, 2010*). They unveil a novel molecular mechanism that integrates pre-TCR signaling with costimulatory Notch signaling to specifically restrict DN3 cell differentiation uncoupled from proliferation and survival. Broad expression of IP$_3$ 3-kinases and PI3Ks, IP$_4$ detection in multiple tissues (*Sauer and Cooke, 2010*) and common PI3K implication in costimulation raise the possibility that 'metabokinetic' control and costimulation-enforcement through non-canonical PI3K antagonism by IP$_3$ 3-kinases are broadly relevant.

## Materials and methods

### Mice

Our C57BL/6 *Itpkb*$^{-/-}$ mice were described in (*Sauer et al., 2013*). All animal studies were approved by the Scripps Research Institute animal care and use committee and conform to all relevant regulatory standards. Mixed bone marrow chimeras were generated as in (*Sauer et al., 2013*). For *in vivo* induction of *Rag2*$^{-/-}$ DN3 cell differentiation, 10 μg anti-CD3 antibodies (BD Biosciences, San Jose, CA, clone 145-2C11) were injected *i.p.* 1–3 days later, the mice were euthanized and analyzed. Where indicated, the mice were treated orally once daily with 10 mg/kg LY-411,575 (*Wong et al., 2004*) or vehicle (5% polyethylene glycol, 3% propylene glycol, 1% ethanol, 0.4% methylcellulose). The first dose was administrated 3–4 hr prior to α-CD3 injection.

For BrdU incorporation assays, we injected mice *i.p.* with 100 μl BrdU [10 mg/ml] and analyzed thymi 4 hr later. For preparation of thymocyte suspensions, thymi were placed in M199 medium/2% FCS/1x penicillin/streptomycin/glutamate at room temperature and single-cell suspensions prepared by passage through a 40 μm mesh (BD Biosciences).

### Flow cytometry and cell sorting

Thymocytes were stained with fluorochome-conjugated antibodies against CD2 (clone RM2-5), CD3 (145-2C11, eBiosciences), CD4 (GK1.5), CD5 (53–7.3), CD8α (53–6.7), CD24/HSA (M1/69), CD25 (3C7), CD27 (LG.3A10), CD44 (IM7), CD71 (R17217), CD98 (RL388, eBiosciences), CD127 (A7R34), TCRβ (H57-597) or CXCR4 (2B11, eBiosciences). Our lineage (Lin) cocktail included biotinylated antibodies against CD11b (M1/70), CD11c (N418), CD19 (6D5), B220 (30-F11), CD49b (DX5), Gr-1 (RB6-8C5), Ly-76/Ter119 (TER-119) and TCRγ/δ (GL3). Unless indicated otherwise, all antibodies were from Biolegend. For intracellular staining, cells were permeabilized with BD Cytofix/Cytoperm kits or 0.3% Triton X100 and stained with antibodies against Ki-67 (B56, BD Biosciences), phospho-Akt T$_{308}$

(C31E5E, Cell Signaling Technology), phospho-mTORC1 $S_{2481}$ (poly6517, Biolegend) (*Soliman et al., 2010*), phospho-ribosomal protein S6 $S_{235}/S_{236}$ (D57.2.2E, Cell Signaling Technology), Glut1 (Fitzgerald Industries), phospho-Erk $T_{202}/Y_{204}$ (D13.14.4E, Cell Signaling Technology), Akt (Cell Signaling Technology) or isotype controls (Cell Signaling Technology) followed by anti-rabbit IgG secondary antibodies (Cell Signaling Technology) if needed. Calyculin A (Cell Signaling) was used according to the manufacturer's protocol. Annexin V staining was performed using eBiosciences Annexin V apoptosis detection kits. BrdU incorporation was assayed with BD Biosciences BrdU-FITC kits, using 0.8 μl anti-BrdU-FITC antibodies per $10^6$ cells. All data were acquired on an LSRII flow cytometer (BD Biosciences) and analyzed using *FlowJo* software.

For DN3 cell purification, thymocytes were first immunomagnetically depleted of CD3, CD4 or CD8α positive cells using biotinylated antibodies against CD3 (145-2C11, Biolegend), CD4 (GK1.5, eBiosciences) and CD8α (53–6.7, eBiosciences), anti-biotin microbeads and LS columns (Miltenyi Biotec) following the manufacturer's protocol, and then stained with anti-CD44-FITC (IM7, eBiosciences), anti-CD25-PerCp-Cy5.5 (3C7, Biolegend) and Streptavidin-APC (Life Technologies). CD44⁻CD25⁺SA⁻ DN3 cells were sorted on a BD FACS Aria cell sorter. CD53⁻ DP thymocytes were purified as in (*Huang et al., 2007*).

## Fetal thymic organ cultures

Different lobes from the same thymus of an embryonic day 15.5 (E15.5) *Itpkb*$^{+/+}$ or *Itpkb*$^{-/-}$ embryo were cultured on gelfoam sponges in complete DMEM-10 with vehicle (100% ethanol) or 20 μM rapamycin in ethanol (BIOTANG/TSZCHEM). New vehicle or rapamycin were added on culture days 1, 2 and 3. The lobes were analyzed on day 4.

## Op9 cell co-cultures

OP9 or OP9DL1 cells (*Ciofani and Zuniga-Pflucker, 2005*; *Ciofani et al., 2004*) were seeded at 8000 cells per well and incubated for 24 hr in OP9 Culture Media (alpha-MEM, Life Technology/15% FCS/1x Penicillin and Streptomycin), followed by addition of 70,000–100,000 sorted DN3 cells per well together with 1 ng/ml recombinant mouse IL-7 (PeproTech) and once on day 0 (rapamycin, Akt-I) or once-daily (2DG) addition of carrier, 500 nM Akt-inhibitor VIII (*Sauer et al., 2013*) (Akt-I, Calbiochem) in DMSO, 4 μM rapamycin (BIOTANG/TSZCHEM) in ethanol or 500 μM 2-deoxy-D-glucose (*Wang et al., 2011*) (2DG, SIGMA) in PBS (all concentrations final).

## Biochemistry

Thymocytes were lysed as previously described (*Huang et al., 2007*) in 1% Triton X-100/60 mM octylglucoside/150 mM NaCl/25 mM Tris-HCl, pH 7.5/1 mM EDTA containing Roche Complete Mini Protease Inhibitor and PhosSTOP Phosphatase Inhibitor Cocktails. Lysates were incubated for 20 min at 4°C, then cleared by centrifugation at 14,000 g for 10 min at 4°C. For immunoprecipitations, pre-cleared lysates were incubated for 1.5 hr with anti-Itpkb antibodies (G-20, Santa-Cruz Biotechnology) followed by incubation with Protein G-conjugated beads for 1.5 hr. Beads were washed 3 times with 1x lysis buffer, denatured in 1x sample buffer at 99°C for 10 min and analyzed via SDS-PAGE/immunoblot. For immunoblot analysis, nitrocellulose membranes were incubated overnight at 4°C with anti-Itpkb (#AP8167b, Abgent) or anti-PLCγ1 (#2822, Cell Signaling Technology) antibodies and then for 45 min with anti-rabbit-HRP secondary antibodies (Bio-Rad Laboratories) in TBS. Bound antibodies were detected by enhanced chemiluminescence (ECL kit, GE Healthcare).

## Mathematical modeling

The kinetics of DN thymocyte differentiation were modeled by a set of linear ordinary differential equations (ODE). In these, the rate constants for DN3 cell generation and for thymocyte subset turnover were similar between genotypes. The rate constants for DN3-to-DP cell differentiation were increased over two2–fold for *Itpkb*$^{-/-}$ cells. The ODE were solved by pen and paper calculations and results verified using BIONETGEN.

## Modeling steady-state thymocyte developmental kinetics in WT vs. *Itpkb*$^{-/-}$ mice

We have built a linear ordinary differential equation (ODE) based model for the kinetics of β-selection in the presence or absence of Itpkb. We approximated the arrival of hematopoietic progenitors followed by their successive maturation via DN1 to DN3 cells as a constant influx rate $K$ (*Figure 5A*). DN3 cells transit to the DN4 stage with a differentiation rate $K_1$ or turn over at a turnover rate $K_{d1}$. Similarly, cells at the DN4 stage can further differentiate into ISP at a rate $K_2$ or turn over at a rate $K_{d2}$. ISP cells further mature into DP cells at a rate $K_3$, or turn over a rate $K_{d3}$. Most DP cells turn over through death by neglect at a rate $K_{d4}$. Only very few DP cells mature to SP thymocytes (*Starr et al., 2003*); these are ignored. In the ODE, we denote the numbers of DN3, DN4, ISP and DP cells as $C_1$, $C_2$, $C_3$ and $C_4$, respectively.

$$\frac{dC_1}{dt} = K - K_1 C_1 - K_{d1} C_1$$

$$\frac{dC_2}{dt} = K_1 C_1 - K_2 C_2 - K_{d2} C_2$$

$$\frac{dC_3}{dt} = K_2 C_2 - K_3 C_3 - K_{d3} C_3 \tag{S1}$$

$$\frac{dC_4}{dt} = K_3 C_3 - K_{d4} C_4$$

It is reasonable to assume that the thymocyte subset population sizes represent steady state solutions of the above ODEs. The ODEs reach the steady state at time scales much longer than the times associated with the kinetic rate constants. At the steady state, $dC_1/dt=dC_2/dt=dC_3/dt=dC_4/dt=0$. This implies that the influx of cells to a particular stage due to differentiation is balanced by outflux due to both differentiation into the next stage and cell death. The steady state solutions of the ODEs are then given by

$$C_1 = \frac{K}{K_1 + K_{d1}}$$

$$C_2 = \frac{K K_1}{(K_1 + K_{d1})(K_2 + K_{d2})}$$

$$C_3 = \frac{K K_1 K_2}{(K_1 + K_{d1})(K_2 + K_{d2})(K_3 + K_{d3})} \tag{S2}$$

$$C_4 = \frac{K K_1 K_2 K_3}{K_{d4}(K_1 + K_{d1})(K_2 + K_{d2})(K_3 + K_{d3})}$$

This implies that regardless of the initial values of the population sizes, the system will reach the above concentrations at long times. DN4 cells and ISP are highly proliferative compared to DN3 and DP cells but show similar overall viability as those, resulting in lower turnover rates (*Figure 3C,D*). The DN4 cell and ISP cell turnover rates ($K_{d2}$, $K_{d3}$) are thus small compared to the respective differentiation rates ($K_2$, $K_3$), such that $K_2/(K_{d2}+K_2) \approx 1$ and $K_3/(K_{d3}+K_3) \approx 1$. In this case, equation (*S2*) can be further simplified as

$$C_1 \approx \frac{K}{K_1 + K_{d1}}$$

$$C_2 \approx \frac{K K_1}{K_2(K_1 + K_{d1})}$$

$$C_3 \approx \frac{K K_1}{K_3(K_1 + K_{d1})} \tag{S3}$$

$$C_4 \approx \frac{K K_1}{K_{d4}(K_1 + K_{d1})}$$

From equation (*S3*), we observe that when differentiation rates $K_1$ (DN3 to DN4 cells), $K_2$ (DN4 cells to ISP) and $K_3$ (ISP to DP cells) increase while progenitor influx rate K is constant, then the

numbers of DN3 cells ($C_1$) and DP cells ($C_4$) remain relatively unaffected, while the numbers of DN4 cells ($C_2$) and ISP ($C_3$) decrease. Indeed, increasing $K_1$, $K_2$ and $K_3$ to the values in *Table 1* modeled the steady state DN cell subset number distribution in WT vs. *Itpkb*-/- (increased $K_{1/2/3}$) mice (*Figure 2C*, *Figure 5B*).

## Modeling thymocyte developmental kinetics in $\alpha$-CD3 antibody injected *Rag2*-/- mice

In *Rag2*-/-*Itpkb*+/+ or *Rag2*-/-*Itpkb*-/- mice, essentially all thymocytes are arrested at the DN3 stage. $\alpha$-CD3 antibody injection triggers their synchronized maturation to DP cells with high proliferation and viability of all developmental stages (*Figure 6C–H*). Progenitor influx within 3 days is negligible. Therefore, we set the progenitor influx rate ($K$) to 0. We started our calculation with C(0) = 3 × $10^8$ DN3 cells and infinitesimally low numbers of DN4, ISP and DP cells, respectively. Otherwise, we used the same differentiation rates and turnover rates as above (*Table 1*). The kinetics of the thymocyte population sizes can then be described by

$$\frac{dC_1}{dt} = -K_1 C_1 - K_{d1} C_1$$

$$\frac{dC_2}{dt} = K_1 C_1 - K_2 C_2$$

$$\frac{dC_3}{dt} = K_2 C_2 - K_3 C_3 \tag{S4}$$

$$\frac{dC_4}{dt} = K_3 C_3 - K_{d4} C_4$$

Equation (*S4*) is solved analytically. To express the solution compactly, we use the variable $\tilde{K}_1 = K_1 + K_{d1}$:

$$C_1(t) = C_1(0)\exp(-\tilde{K}_1 t)$$

$$C_2(t) = \frac{K_1 C_1(0)}{K_2 - \tilde{K}_1}[\exp(-\tilde{K}_1 t) - \exp(-K_2 t)]$$

$$C_3(t) = \frac{K_1 K_2 C_1(0)}{K_2 - \tilde{K}_1}\left[\frac{\exp(-\tilde{K}_1 t)}{K_3 - \tilde{K}_1} - \frac{\exp(-K_2 t)}{K_3 - K_2} + \frac{(K_2 - \tilde{K}_1)\exp(-K_3 t)}{(K_3 - \tilde{K}_1)(K_3 - K_2)}\right]$$

$$C_4(t) = \frac{K_1 K_2 K_3 C_1(0)}{K_2 - \tilde{K}_1}\left[\frac{\exp(-\tilde{K}_1 t)}{(K_{d4} - \tilde{K}_1)(K_3 - \tilde{K}_1)} - \frac{\exp(-K_2 t)}{(K_3 - K_2)(K_{d4} - K_2)} + \frac{(K_2 - \tilde{K}_1)\exp(-K_3 t)}{(K_3 - \tilde{K}_1)(K_3 - K_2)(K_{d4} - K_3)}\right]$$

$$- \frac{K_1 K_2 K_3 C_1(0)}{K_2 - \tilde{K}_1}\left[\frac{1}{(K_{d4} - \tilde{K}_1)(K_3 - \tilde{K}_1)} - \frac{1}{(K_3 - K_2)(K_{d4} - K_2)} + \frac{(K_2 - \tilde{K}_1)}{(K_3 - \tilde{K}_1)(K_3 - K_2)(K_{d4} - K_3)}\right]\exp(-K_{d4} t)$$

(S5)

In equation (*S5*), t denotes the time post $\alpha$-CD3 antibody injection in hr. The ODE were solved by pen and spaper calculations and results verified using BIONETGEN (*Hlavacek et al., 2006*). Results of a simulation with 3 hr time increments for 3 days are shown in *Figure 6A,B*.

**Table 1.** Parameters used in our mathematical models.

| Genotype | K (cells/day) | K₁ (days⁻¹) | K_d1 (days⁻¹) | K₂ (days⁻¹) | K₃ (days⁻¹) | K_d4 (days⁻¹) |
|---|---|---|---|---|---|---|
| WT | 15.4 × $10^4$ | 0.1 | 0.1 | 0.162 | 0.07 | 0.00058 |
| Itpkb-/- | 15.4 × $10^4$ | 0.2 | 0.1 | 0.486 | 0.21 | 0.00058 |
| Rag2-/-Itpkb+/+ | 0 | 0.1 | 0.1 | 0.162 | 0.07 | 0.00058 |
| Rag2-/-Itpkb-/- | 0 | 0.2 | 0.1 | 0.486 | 0.21 | 0.00058 |

## Statistical analyses

Aggregated results are shown as mean ± SEM. p values for the indicated comparisons were calculated by two-tailed unpaired Student's t-test. Group sizes are described in the figure legends. Significant p values are denoted by asterisks: $*p<0.05$; $**p<0.01$; $***p<0.001$; $****p<0.0001$. All statistical analyses were performed in *Prism*.

## Acknowledgements

We thank Boreth Eam for technical help, the TSRI vivarium for mouse husbandry, Kris Hogquist for the *Nr4a1/Nur77-GFP* transgenic mice, Juan-Carlos Zúñiga-Pflücker for the OP9 and OP9-DL1 cells, Wendy Havran and Deborah Witherden for help with FTOC, Nick Gascoigne and Changchun Xiao for critical reading of the manuscript.

## Additional information

### Funding

| Funder | Grant reference number | Author |
| --- | --- | --- |
| National Institutes of Health | AI007606 | Luise Westernberg |
| National Institutes of Health | AI089805 | Yina Hsing Huang |
| American Association of Immunologists | | Stephanie Rigaud |
| Deutsche Forschungsgemeinschaft | SI 1547/1-1 to S.S | Sabine Siegemund |
| National Institutes of Health | AI108880 | Jayajit Das |
| National Institutes of Health | AI090115 | Jayajit Das |
| National Institutes of Health | AI070845 | Karsten Sauer |
| The Leukemia and Lymphoma Society | 1440-11 | Karsten Sauer |
| National Institutes of Health | GM100785 | Karsten Sauer |

The funders had no role in study design, data collection and interpretation, or the decision to submit the work for publication.

### Author contributions

LW, SM, JD, Conception and design, Acquisition of data, Analysis and interpretation of data, Drafting or revising the article; CC, YHH, SR, Acquisition of data, Analysis and interpretation of data, Drafting or revising the article; YD, SS, LAN, Acquisition of data, Analysis and interpretation of data; KS, Conception and design, Analysis and interpretation of data, Drafting or revising the article

### Ethics

Animal experimentation: All animal studies were approved by The Scripps Research Institute animal care and use committee (IACUC protocol 06-0355) and conform to all relevant regulatory standards.

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
