## [Decision Letter]

Thank you for submitting your work entitled "Non-canonical PI3K Antagonism by the Kinase Itpkb Delays Thymocyte β-Selection and Renders it Notch-Dependent" for peer review at *eLife*. Your submission has been favorably evaluated by Tadatsugu Taniguchi (Senior editor) and three reviewers, one of whom is a member of our Board of Reviewing Editors.

The reviewers have discussed the reviews with one another and the Reviewing editor has drafted this decision to help you prepare a revised submission.

Summary:

This work by Sauer and colleagues provides an insightful analysis showing how counter-regulation of PI3K signalling by Itpkb in precursor CD4^-^CD8^-^ thymocytes sets up the need for Notch signals to provide an enabling signal to allow the pre-TCR to induce successful differentiation of these cells to the CD4^+^CD8^+^ DP stage of T cell development. The authors make use of Itpkb-deficient mice in several differentiation models and their findings strongly support their conclusions, deepening our understanding of how preTCR and Notch signals collaborate to support thymocyte differentiation at the TCR-β selection checkpoint. The data provide interesting lessons regarding molecular strategies for integration of signaling via multiple developmental cues, which is an issue of broad biological interest. There are a few concerns, mainly regarding the interpretations and implications of the data, that it would be advisable to address prior to publication.

Essential revisions:

1) In Figure 4, the authors generate mixed bone marrow chimeras to test whether the altered β-selection is due to thymocyte-intrinsic Itpkb-loss. Although most of the data recapitulate the defect seen in Itpkb-deficient mice, there are mature (HSA^low^TCR^high^CD4^-^CD8^+^) CD8 T cells in the chimeras, a population essentially non-existent in Itpkb-deficient mice. It would be useful to have the provenance of these cells clarified and the implications discussed.

2) The data show some residual DP stage generation from WT thymocytes even on parental OP9 cells not expressing Notch ligand (Figure 7). Thus, additional data showing DP transition of Itpkb-deficient thymocytes even when Notch signalling is actively blocked would be useful to support the conclusion that Itpkb-loss causes Notch-independent development to the DP stage.

3) The authors use the ability of inhibitors of Akt, mTOR and glucose metabolism to reduce developmental transition in Itpkb-deficient thymocytes, coupled with higher levels of activated Akt, mTOR and S6, to argue that the hyper-responsiveness and accelerated developmental transition in these Itpkb-deficient thymocytes is MEDIATED by these pathways. However, the data only provide evidence that this functional phenotype in the Itpkb-deficient genotype remains *sensitive* to these pathways, since inhibition of these pathways apparently decreases DP transition to a similar in both WT and Itpkb-deficient thymocytes. In fact, such treatment does not completely reverse the accelerated DN3-to-DP transition of Itpkb-deficient thymocytes (Figure 7, Figure 9), raising the possibility that pre-TCR-induced Akt/mTOR and metabolic activation may not be the only cause of the altered β-selection seen in Itpkb-deficient mice. In order to claim a specific causal connection, it may be necessary, for example, to show that sub-optimal concentrations of these pathway inhibitors do not affect WT transition but do still affect Itpkb-deficient cell transition, coupled to a demonstration that inhibition of other pathways (important for developmental transition but not modulated by Itpkb) does not show such a differential effect on WT versus Itpkb-deficient thymocytes. In the absence of such evidence, it may be more appropriate to discuss these various possibilities.

4) It is not clear if the authors make distinctions between 'kinetic' acceleration of developmental transition and enhanced sensitivity to developmental cues. Thus, are pre-TCR-mediated activation 'thresholds' lower in Itpkb-deficient thymocytes, and/or, do they show an actual increase in the rates of cellular events mediating as well as resulting from pre-TCR-mediated activation signals? Which data provide evidence for one, and which for the other, possibility? How does 'more' signal, as indicated by higher levels of inducible gene products such as Nur77, translate into kinetically faster developmental throughput? It would be interesting if the authors could discuss these issues.

---

## [Author Response]

Essential revisions: 1) In Figure 4, the authors generate mixed bone marrow chimeras to test whether the altered β-selection is due to thymocyte-intrinsic Itpkb-loss. Although most of the data recapitulate the defect seen in Itpkb-deficient mice, there are mature (HSA^low^TCR^high^CD4^-^CD8^+^) CD8 T cells in the chimeras, a population essentially non-existent in Itpkb-deficient mice. It would be useful to have the provenance of these cells clarified and the implications discussed.

We thank the reviewers for pointing this out. Double-checking the original FACS data, we found that the HSA/TCRβ plot previously shown for the *Itpkb^-/-^* compartment was wrong. We apologize for this mistake, which had been unnoticed during proofreading. Revised Figure 4 now shows the correct plot. The plot shown for the *Itpkb^+/+^* compartment was the correct one and has remained unchanged. The correct plot shows 4% mature CD8 cells among the <0.5% total CD8 cells in the *ItpkB^-/-^* compartment (0.02% total), contrasting with 79% mature CD8 cells among the 3% CD8 cells in the *wt* compartment (2.37% total). These values are in the same order of magnitude as those for *Itpkb^-/-^* mice in Figure 2 and as we have previously published for *Itpkb^-/-^* mice and the *Itpkb^-/-^* compartment of BM chimeras (Wen et al., 2004, Huang et al., 2007). Thus, the corrected BM chimera data are consistent with the data in *Itpkb^-/-^* mice and the literature. Several additional BM chimeras confirmed these results.

*2) The data show some residual DP stage generation from WT thymocytes even on parental OP9 cells not expressing Notch ligand (*Figure

*7A). Thus, additional data showing DP transition of Itpkb-deficient thymocytes even when Notch signalling is actively blocked would be useful to support the conclusion that Itpkb-loss causes Notch-independent development to the DP stage.*

To address this concern in an orthogonal approach to the OP9-system, we analyzed DN3 cell maturation in *Itpkb^-/-^Rag2^-/-^* versus *Rag2^-/-^* mice treated with vehicle or the γ-secretase/Notch inhibitor LY-411,575 (Wong et al., 2004) post α-CD3 injection. We found that LY-411,575 strongly impaired the α-CD3 induced DN cell development into ISP and DP cells in *Rag2^-/-^* but not *Itpkb^-/-^Rag2^-/-^* mice (new Figure 10). This suggests that *Itpkb*-loss reduces the Notch-dependence of DN thymocyte development to DP cells not only in the OP9 system, but alsoin vivo. We added a paragraph at the end of the Results and text to the first paragraph of the Methods to describe this experiment and the results.

*3) The authors use the ability of inhibitors of Akt, mTOR and glucose metabolism to reduce developmental transition in Itpkb-deficient thymocytes, coupled with higher levels of activated Akt, mTOR and S6, to argue that the hyper-responsiveness and accelerated developmental transition in these Itpkb-deficient thymocytes is MEDIATED by these pathways. However, the data only provide evidence that this functional phenotype in the Itpkb-deficient genotype remains sensitive to these pathways, since inhibition of these pathways apparently decreases DP transition to a similar in both WT and Itpkb-deficient thymocytes. In fact, such treatment does not completely reverse the accelerated DN3-to-DP transition of Itpkb-deficient thymocytes (Figure 7, Figure 9), raising the possibility that pre-TCR-induced Akt/mTOR and metabolic activation may not be the only cause of the altered* β-*selection seen in Itpkb-deficient mice. In order to claim a specific causal connection, it may be necessary, for example, to show that sub-optimal concentrations of these pathway inhibitors do not affect WT transition but do still affect Itpkb-deficient cell transition, coupled to a demonstration that inhibition of other pathways (important for developmental transition but not modulated by Itpkb) does not show such a differential effect on WT versus Itpkb-deficient thymocytes. In the absence of such evidence, it may be more appropriate to discuss these various possibilities*.

As proposed by the reviewers, we now discuss these possibilities in the fifth paragraph of the Discussion. We also weakened the statements about causative Akt/mTOR hyperactivation in the first paragraph of the subsection “Itpkb restricts pre-TCR signaling to delay β-selection and render it Notch-dependent” and in the paragraph of the Discussion mentioned above.

4) It is not clear if the authors make distinctions between 'kinetic' acceleration of developmental transition and enhanced sensitivity to developmental cues. Thus, are pre-TCR-mediated activation 'thresholds' lower in Itpkb-deficient thymocytes, and/or, do they show an actual increase in the rates of cellular events mediating as well as resulting from pre-TCR-mediated activation signals? Which data provide evidence for one, and which for the other, possibility? How does 'more' signal, as indicated by higher levels of inducible gene products such as Nur77, translate into kinetically faster developmental throughput? It would be interesting if the authors could discuss these issues.

We added the third paragraph to the Discussion to discuss these interesting issues.